# Sea-level rise in a coastal marsh: linking increasing tidal inundation, decreasing soil strength and increasing pond expansion

Mona Huyzentruyt[1*], Lennert Schepers[1], Matthew L. Kirwan[2], Glenn R. Guntenspergen[3], Stijn Temmerman[1]

[1]ECOSPHERE research group, University of Antwerp, Antwerp, Belgium

[2]Virginia Institute for Marine Science, Williams and Mary, Gloucester Point, Virginia, USA

[3]U.S. Geological Survey, Eastern Ecological Science Center, Duluth, MN, USA

*Correspondence to*: Mona Huyzentruyt (mona.huyzentruyt@uantwerpen.be)

**Abstract.** Coastal marsh conversion into ponds, which may be triggered by sea-level rise, is considered an important driver of marsh loss and their valuable ecosystem services. Previous studies have focused on the role of wind waves in driving the expansion of interior marsh ponds, through lateral erosion of marsh edges surrounding the ponds. Here, we propose another mechanism between sea-level rise, increasing marsh inundation, and decreasing marsh soil strength (approximated here as resistance to shear and penetration stress), that further contributes to marsh erosion and pond expansion. Our field measurements in the Blackwater marshes (Maryland, USA), a microtidal marsh system with organic-rich soils, indicate that (1) an increase in tidal inundation time of the marsh surface above a certain threshold (around 50% of the time) is associated with a substantial loss of strength of the surficial soils; and (2) this decrease in soil strength is strongly related to the amount of belowground vegetation biomass, which is also found to decrease with increasing tidal inundation at pond bottoms, where the soil has a very low strength. Our finding of decreasing marsh soil strength along a spatial gradient of increasing marsh inundation coincides with a gradient of increasing historical marsh loss by pond expansion, suggesting that feedbacks between sea-level rise, increasing marsh inundation and decreasing marsh soil strength combine to amplify marsh erosion and pond expansion.

**Graphical abstract.**

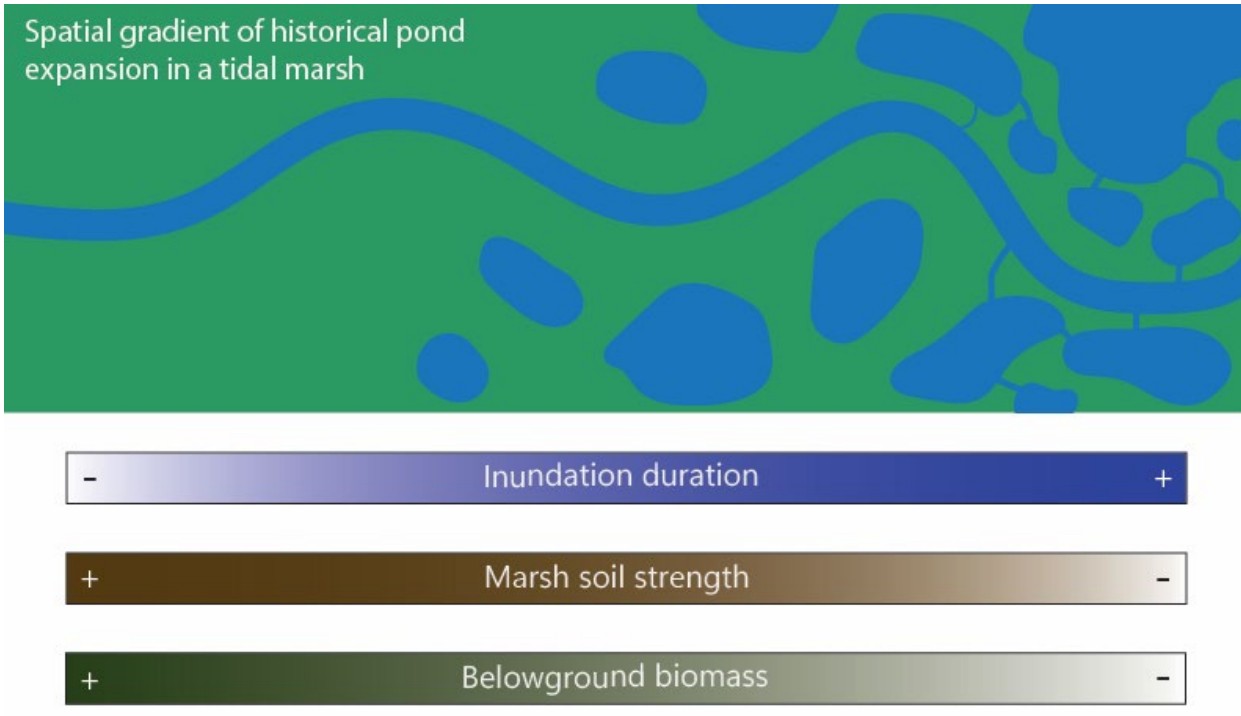

## 1 Introduction

Vegetated tidal marshes provide highly valued ecosystem services, including nature-based climate mitigation by carbon
sequestration (Duarte et al., 2013; Macreadie et al., 2019; McLeod et al., 2011; Temmink et al., 2022), nature-based shoreline
protection by attenuating storm waves and storm surges (Möller et al., 2014; Schoutens et al., 2019; Stark et al., 2015;
Temmerman et al., 2023; Zhu et al., 2020), and providing  nursery grounds for marine fisheries (Barbier et al., 2011). However,
tidal marshes and their ecosystem services are vulnerable to degradation through various mechanisms. One widely considered
threat is sea-level rise, which results in increasing tidal inundation, may trigger vegetation die-off and cause pond formation
within marshes, in situations where sediment accretion is insufficient to allow marshes to build up their soil surface elevation
with the rising sea-level (Coleman et al., 2022; Kirwan et al., 2016; Mariotti, 2016; Ortiz et al., 2017; Schepers et al., 2017;
Vinent et al., 2021).

Previous studies on pond formation and lateral pond expansion mostly focused on the role of waves in driving the lateral
erosion of the marsh edges surrounding the interior marsh ponds (Mariotti, 2016; Morton et al., 2003; Ortiz et al., 2017;
Penland et al., 2000). Aerial image analyses have shown that lateral erosion rates of the marsh edges accelerate when ponds
exceed a critical threshold length of about 200 to 1000 m (Mariotti, 2016; Ortiz et al., 2017). Further, field observations have

demonstrated that ponds with larger length tend to be deeper (Schepers et al., 2020a). Models suggest this is attributed to a positive feedback between the pond length, wind fetch length, wave heights generated on the ponds, and hence wave-induced erosion of pond bottoms and pond edges. This creates a feedback that may give rise to run-away pond enlargement and marsh loss, especially where tidal range and sediment supply are low (Mariotti, 2020; Vinent et al., 2021). Relatively little is known on the processes driving the expansion of interior marsh ponds before they reach this critical threshold size, but a number of studies indicate that biogeochemical processes are at play, such as sulphate reduction in early ponds leading to decomposition of soil organic matter and hence further pond deepening (Spivak et al., 2018; van Huissteden & van de Plassche, 1998a) and production of phytotoxic substances in soil pore water, such as sulfides and ammonium along the marsh edges surrounding ponds, which may trigger vegetation die-off and pond enlargement (Himmelstein et al., 2021).

However, there is a paucity of empirical knowledge examining the role of potential feedbacks between sea-level rise and marsh soil strength in affecting the process of lateral marsh erosion and pond expansion. The soil strength of marshes is known to influence lateral erosion rates (Valentine & Mariotti, 2019), and in this paper, we investigate the hypothesis that the marsh soil strength (measured as resistance against shear and penetration stress) is decreasing with increasing tidal inundation of marshes, which may trigger a positive feedback between sea-level rise, increasing marsh inundation, lower soil strength and higher vulnerability to lateral marsh erosion and pond expansion. The strength of marsh soils is known to depend on sediment properties and belowground plant biomass structure (Chen et al., 2012; Coops et al., 1996; Feagin et al., 2009a; Francalanci et al., 2013; Stoorvogel, de Smit, et al., 2025; Stoorvogel et al., 2024; Wang et al., 2017). Furthermore, a few experimental studies have demonstrated the effect of increased inundation on belowground biomass production and decomposition. Kirwan and Guntenspergen (2012, 2015) found in field mesocosm experiments that a small increase in the hydroperiod (i.e., the percentage of time the marsh is inundated by the tides) from values less than or equal to 35-45 % initially stimulates belowground plant growth, but productivity quickly declines once the hydroperiod exceeds 35-45 %. This decline of belowground productivity above a hydroperiod threshold has been confirmed by other field mesocosm experiments and is supposed to be related to increased plant stress in response to an increasing tidal hydroperiod (Langley et al., 2013; Snedden et al., 2015; Voss et al., 2013; Watson et al., 2014). Decomposition rates of soil organic matter appear to be rather constant and relatively unaffected by inundation (Kirwan et al., 2013a; Mueller et al., 2016). Hence, these mesocosm experiments suggest that increasing inundation can decrease belowground productivity of tidal marsh vegetation. Here, we hypothesize that the latter can further affect the marsh soil strength. However, apart from two studies documenting weak soil strengths in degrading coastal marshes in the Mississippi delta (Day et al., 2011; Howes et al., 2010), we are only aware of one study linking spatial variations in marsh soil strength in relation to a field gradient of increasing marsh hydroperiod (Jafari et al., 2024). This relationship was however quantified in a marsh system without signs of degradation as a result of sea-level rise, hence, it remains poorly understood if there are potential feedbacks between sea-level rise, marsh soil strength, and marsh loss by lateral erosion and expansion of ponds.

In this study, we quantified and analyzed the changes in soil strength along a well-documented gradient of increasing marsh loss by pond expansion (Schepers et al., 2017) in the organogenic, microtidal Blackwater marshes (Maryland, USA). Our analysis suggests relationships between increasing tidal hydroperiod, decreasing soil strength, and decreasing belowground biomass along the marsh loss gradient, suggesting that decreasing marsh soil strength in response to sea-level rise may amplify marsh erosion and may contribute to runaway marsh collapse.

## 80 2 Methods

### 2.1 Study area

The Blackwater River marshes (Maryland, USA: 38°24' N, 76°40' W, Fig. 1) are microtidal, brackish marshes bordered in the southeast by Fishing Bay, a coastal embayment connected to the Chesapeake Bay (Fig. 1b). Long-term salinity of marsh soil pore water is around 10 to 12 (Kirwan et al., 2013b) but the salinity might change substantialy on seasonal timescales
(Fleming et al., 2011). The mean tidal range decreases from 63 cm at Fishing Bay (bottom right of Fig. 1a) to 6 cm at Lake Blackwater (top left of Fig. 1a) (Fig. 1a; Schepers et al., 2020b). The marshes are characterized by mesohaline marsh vegetation: *Spartina cynosuroides* (L.) Roth is dominant in the marsh zones directly adjacent to the river and the bigger tidal channels. *Spartina alterniflora* Loisel. and *Schoenoplectus americanus* (Pers.) are most abundant in the other areas, often in assemblages with *Spartina patens* Roth and *Distichlis spicata* (L.) Greene (Schepers et al., 2020b).

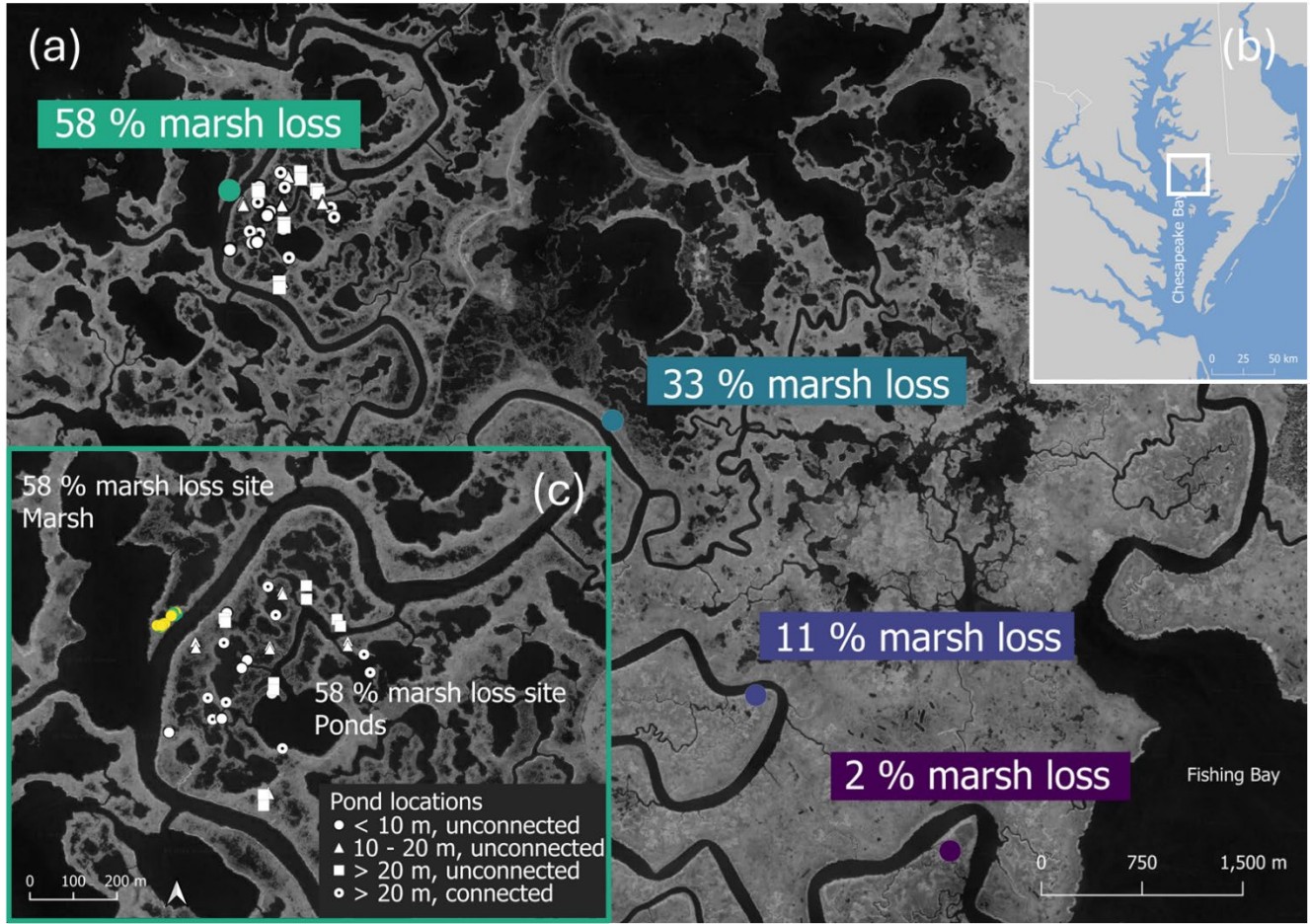

**Figure 1: (a): Aerial images of the Blackwater marshes (black: water, light grey: marsh) with sampling locations (Copernicus – Sentinel data [2025]. Retrieved from © Google Earth Engine, processed by ESA). The marsh loss (i.e. proportion of shallow open water ponds to total marsh area) is quantified for each site based on Schepers et al. (2017). (b) Inset map showing the location of the Blackwater marshes in the Chesapeake Bay. The green box is the extent of panel c. (c): pond locations (white) sampled at site 4. Values in the legend of (c) refer to the average pond diameter in each category. The arrow on the bottom is a North arrow.**

More than 2000 ha of marshland in the Blackwater National Wildlife Refuge have been converted from vegetated marsh to shallow open water ponds since the 1930s (Cahoon et al., 2010). There is a spatial gradient of increasing marsh loss in the upstream direction along the Blackwater River, from intact marshes close to Fishing Bay (southeastern corner on Fig. 1A) to complete marsh loss at Lake Blackwater (northwestern corner of Fig. 1A). Lake Blackwater is now a vast open water area that once consisted of expansive marshes observed in historical aerial photographs (Stevenson et al., 1985; Schepers et al., 2017). Since the 1930s, continuous formation and merging of new ponds has led to the growth of larger bodies of open water and progressive marsh loss (Himmelstein et al., 2021). Spatial patterns across the present-day marsh loss gradient closely resemble the historical, spatio-temporal development of marsh loss of the most degraded areas (Schepers et al., 2017). As a result, the

present-day spatial marsh loss gradient can be considered a chronosequence and marsh loss processes can be studied with space for time substitution (Schepers et al., 2017).

The underlying cause of marsh loss in this area is attributed to insufficient organic and mineral sediment accretion to maintain the surface elevation of marshes in the face of sea-level rise (Ganju et al., 2013; Stevenson et al., 1985). In particular, sediment accretion rates (on average 1.7-3.6 mm $yr^{-1}$ (Stevenson et al., 1985)) are less than the long-term rate of relative sea-level rise of 4.06 mm $yr^{-1}$ in Cambridge, MD, calculated over the period of 1943-2025 (NOAA station 8571892, http://tidesandcurrents.noaa.gov/sltrends, 2025-04-10)). The historical sea level rise rate has been increasing since the 1970's and it has exceeded the sediment accretion rate since the 1990's (NOAA station 8571892, http://tidesandcurrents.noaa.gov/sltrends, 2025-04-10). Moreover, more sediment is exported from the system than imported into it (Ganju et al., 2013). As a result, more than 80 % of marshes in the degraded portions of the study area occupy elevations below the optimum for *Schoenoplectus americanus* productivity (Kirwan & Guntenspergen, 2012). This leads to increased tidal inundation of the vegetation, changes in soil conditions and ultimately marsh vegetation die-off and conversion to shallow open water.

## 2.2 Sampling design

We conducted a field campaign to sample soil cores and to measure soil strength (more detail in paragraph 2.3) from 15 to 24 August 2016. The sampling locations were selected to cover two scales of spatial variability in marsh and pond environments. First, we selected four field sites, with increasing proportion of open water areas to the total marsh area, as a measure of marsh loss rate, more specifically 2 %, 11 %, 33 % and 58 % marsh loss (Fig. 1a, Table 1) (Schepers et al., 2017, 2020a, 2020b). At each field site, we selected five locations with monospecific stands of *Schoenoplectus americanus*. This species was selected because it is the most abundant in marsh zones surrounding existing ponds and hence expected to be most vulnerable to conversion to ponds (Schepers et al., 2020b). Locations located > 20 m from the river and > 1.5 m from ponds were selected to reduce potential edge effects. The five locations at each field site were selected to have soil surface elevations similar to the average marsh surface elevation of each site as measured in our previous studies (Schepers et al. 2017, 2020a).

Second, at the 58 % marsh loss site, we selected additional locations, representing different types of marsh and pond environments that were more locally distributed (Fig. 1c, Table 1). We selected five additional locations within marsh vegetation with a lower surface elevation than the average marsh elevation. We also selected seven locations in small (0.5-5 m²), bare patches surrounded by marsh vegetation. Additionally, we categorized ponds into four pond classes (Fig. 1c), based on average diameter and connection to the tidal channel network: (i) ponds with an (average) diameter of <10 m and not connected to tidal channels; (ii) ponds with 10-20 m (average) diameter and unconnected; (iii) ponds with >20 m (average) diameter and unconnected; and (iv) ponds >20 m (average) diameter and connected to the channel network (Fig. 1c, Table 1). These pond classes correspond to different ages of the ponds, as the ponds of class (i) became visible on aerial images between

1995 and 2010, class (ii) ponds all appear since 1995 images, class (iii) ponds became visible on images taken between 1981 to 1995, and class (iv) ponds on images taken between 1938 and 1981 (Schepers et al., 2017). Five ponds of each category were selected for sampling and for each pond, the north and south side was sampled.

At each of the sampling locations described above (and Fig. 1), the elevation relative to the North American Vertical Datum of 1988 (NAVD88) was recorded with a high-precision GPS (Trimble R10 RTK-GPS, vertical error <1.5 cm). At the ponds, five pond bottom elevations were recorded within 1m along the pond edge to account for possible variability. Making use of tidal water level time series measured at each field site during a previous field campaign (using Hobo U20L-02 sensors; from August 14 to October 29, 2014, Schepers et al. 2020a), we recalculated the surface elevations, originally measured relative to NAVD88, to surface elevations above the local mean sea-level (m amsl) (Table 1). Further, we calculated for each sampling location the duration of tidal inundation (further referred to as the hydroperiod) as the % of time that the water level is higher than the soil surface elevation of the location (Table 1).

## 2.3 Soil strength measurements

Two proxies of soil strength were measured with (1) a shear vane device and (2) a soil penetrologger. These measures represent two different aspects of soil stability. The shear vane (H-4227 Vane Inspection Set, Humboldt Mfg. Co., USA) measures the maximum shear stress (N m$^{-2}$) to break the soil from torsion exerted by a rod fitted with four vanes inserted into the soil and rotated at different depths. The maximum shear stress to break the soil is referred to as the shear vane soil strength (in N m$^{-2}$). At all marsh points (five plots in the 2 %, 11 % and 33 %m marsh loss site and 17 in the 58 % marsh loss site), we measured the shear vane soil strength just below the soil surface (within the rooting zone) and at 30 cm below the soil surface (below the rooting zone). For ponds, we only performed measurements at the surface of the pond bottom. We also examined another aspect of soil strength by measuring the cone penetration resistance (in N m$^{-2}$) with a soil penetrologger (06.15.SA, Eijkelkamp, NL). This device measures resistance to vertical penetration and electronically records the force (N) needed to push a cone with a given surface area through the soil, and simultaneously registers the depth by an ultrasonic sensor. The soil penetration resistance in N/m² was calculated by dividing the force by the cone base area. The measurement was taken at all marsh (five plots in the 2 %, 11 % and 33 % marsh loss site and 17 in the 58 % marsh loss site) a pond sites in the upper 80 cm of sediment. Each soil strength measurement was replicated five times within a radius of 0.5 m from the sampling points.

## 2.4 Belowground biomass sampling and sediment analysis

At the marsh locations (five plots in the 2 %, 11 % and 33 % marsh loss site and 10 in the 58 % marsh loss site), soil cores were collected to a depth of 15 cm with a 10 cm diameter stainless steel coring tube, with a very sharp edge at the bottom of the tube enabling to cut through belowground roots. The upper 15 cm of the pond substrate (which was much more loose material without roots) was sampled with a transparent tube with sharpened edges and vacuum cap. At the bare patches, the loose soil prevented us taking core samples of an exact volume but grab samples of the upper 15 cm were taken for analysis.

170 At each point (five plots in the 2 %, 11 % and 33 % marsh loss site and 17 in the 58 % marsh loss site), two cores were sampled. One of the two cores was dried for minimum 120 h at 105°C to a constant weight to determine dry bulk density. Water content was determined by the difference in weight before and after drying. The other core was sliced in half cores. One half was dried, ground and homogenized with a 0.5-mm grinder (Retsch ZM2000) and heated to 550°C and ashed for four hours to determine the organic content of the soil samples (loss on ignition). The other half of the core was used to determine belowground biomass

175 fractions.

Half cores intended for belowground biomass determination were manually broken apart and thoroughly rinsed with a commercial kitchen spray arm above a sieve with 2 mm maize size to remove all the mineral particles. The rinsed belowground biomass was visually sorted into red rhizomes, white rhizomes, stems and the remaining litter fraction (macro-remains)

180 according to the descriptions in Saunders et al. (2006) (see Appendix A1). The different biomass fractions are characterised by differences in chemical composition (e.g. lignin content and C/N ratio), which has an effect on the decomposition rate (Saunders et al., 2006; Scheffer & Aerts, 2000). Each fraction was dried for minimum 60 h at 70°C to a constant weight. In the bare patches, where we took grab samples, we could not determine an exact volume of the soil samples, but we determined the relative contribution of the different types of belowground biomass.

185 **2.5 Statistical analysis**

The effect of hydroperiod on shear strength and belowground biomass was analysed using linear mixed models (LMM), using field site as a random effect to account for within site clustering. A separate LMM analysis was performed to evaluate the influence of organic matter content, bulk density, water content, hydroperiod and belowground biomass on shear strength, again incorporating field site as a random effect. The differences in bulk density, water content, organic matter, shear strength

190 and belowground biomass between sites were analysed using pairwise Wilcoxon rank sum test with Bonferroni correction. All analyses were executed in R (R core team, 2022), using the lme4 package (Bates et al., 2015) for the linear mixed models. The p-value threshold used is 0.05.

**3 Results**

**3.1 Belowground biomass and marsh soil strength in relation to hydroperiod**

195 The marsh sampling locations were distributed over a gradient in soil surface elevation relative to the local mean sea-level (Table 1). Correspondingly the hydroperiod increased from around 30 % at the sampling locations with highest soil surface elevation relative to mean sea-level to around 90 % at the sampling locations with lowest surface elevation (Table 1, Fig. 2).

**Table 1: Overview of properties of the field sampling locations (Fig. 1): number of samples per location, mean surface elevation (m above local mean sea-level (m amsl)), tidal range (m), and hydroperiod (% of time that a location is inundated by tides). The numbers in the pond location categories refer to the average diameter of the ponds.**

| Sampling location | Vegetation present? | Number of locations (n) | Mean elevation (m amsl) | Hydro-period (%) | Mean tidal range (m) |
|---|---|---|---|---|---|
| Marsh locations: | | | | | |
| 2% marsh loss site | Yes | 5 | 0.35±0.006 | 29.4±0.82 | 0.63 |
| 11 % marsh loss site | Yes | 5 | 0.16±0.007 | 54.3±1.43 | 0.31 |
| 33 % marsh loss site | Yes | 5 | 0.12±0.005 | 58.2±1.60 | 0.20 |
| 58 % marsh loss site | Yes | 5 | 0.11±0.002 | 73.7±0.93 | 0.06 |
| Lower elevation site | Yes | 5 | 0.07±0.014 | 86.5±3.66 | 0.06 |
| Bare patches site | No | 7 | 0.04±0.031 | 91.7±5.29 | 0.06 |
| Pond locations: | | | | | |
| <10 m, unconnected ponds | No | 10 | -0.06±0.027 | 100 | 0.06 |
| 10-20 m, unconnected ponds | No | 10 | -0.08±0.059 | 100 | 0.06 |
| >20 m, unconnected ponds | No | 10 | -0.08±0.068 | 100 | 0.06 |
| >20 m, connected ponds | No | 10 | -0.21±0.115 | 100 | 0.06 |

Even though the regression analysis indicated no significant effect of hydroperiod on belowground biomass nor shear strength (p=0.31 and p=0.24 respectively), our graphs seemed to indicate that the hydroperiod has an influence on the belowground biomass (Fig. 2a) and the shear vane soil strength (Fig. 2b) of the marsh topsoil samples (0-15 cm soil depth). There was an increase in belowground biomass and soil strength from locations at the 2 % marsh loss site (with the lowest hydroperiods around 30 %), to the 11 % marsh loss site (with intermediate hydroperiods around 55 %), followed by a decrease from the 11 % marsh loss site to the lower plots of the 58 % marsh loss site (with highest hydroperiods up to >90 %). For hydroperiods ranging from 55 % up to more than 90 %, the shear vane soil strength of the topsoil decreased systematically with increasing hydroperiod (Fig. 2b). This decrease in marsh soil strength corresponded to the gradient of increasing marsh loss (Fig. 1a, Table 1). The soil bulk density and the soil water content were not significantly different (p=0.28 and p=0.06 respectively) at the different marsh sampling locations (Table 2). The organic matter content is however significantly lower at the bare patches site compared to the 2 % and the 11 % marsh loss site (p<0.05), but not different from the 33 %, the 58 % and the lower elevation sites.

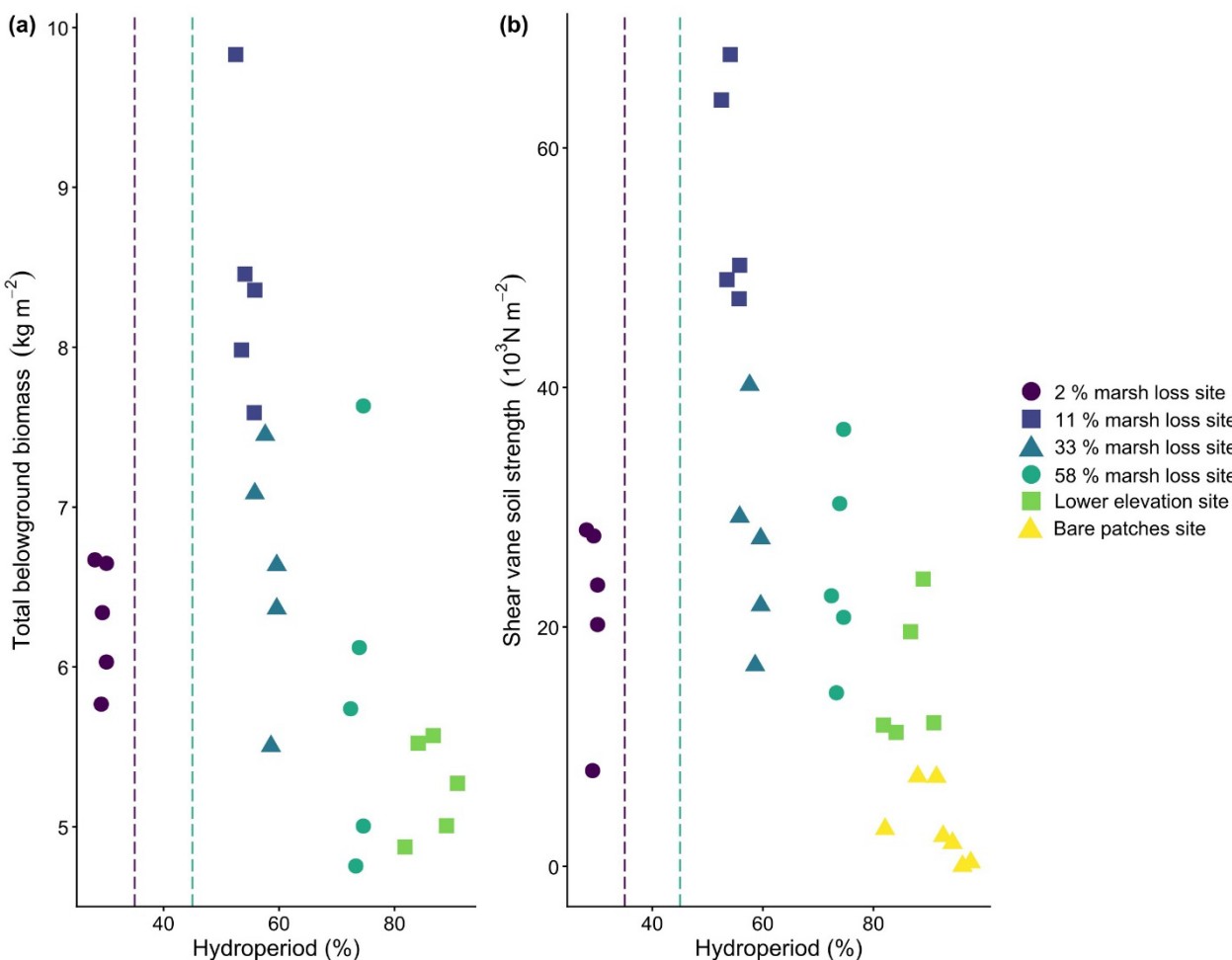

**Figure 2: (a) Total belowground biomass (kg m$^{-2}$ for 0-15 cm soil depth) versus hydroperiod for all vegetated marsh sampling locations (no bare or pond locations). (b) Top-soil shear vane soil strength (10³ N m$^{-2}$, for 0-10 cm soil depth) versus hydroperiod for all vegetated and bare marsh sampling locations (no pond locations). The vertical dashed lines indicate hydroperiods for which belowground biomass production was maximal as determined by an experimental**

**setup close to the 2% and 58% marsh loss sites (Kirwan and Guntenspergen, 2015).**

**Table 2: Overview of organic matter content (%) by loss on ignition, water content (%) and dry bulk density (g cm$^{-3}$) of the topsoil samples (0-15 cm soil depth) at the different sampling locations (see Fig. 1 and Table 1). Average values ± standard deviations. n=5 for vegetated marsh locations, n=7 for bare patches within marshes, n=10 for pond locations. Water content and bulk density could not be measured on bare patches and pond locations. NA indicates variables (water content and dry bulk density) that could not be measured on the pond sediment samples. The numbers in the pond location categories refer to the average diameter of the ponds.**

| Sampling location | Organic matter content (%) | Water content (%) | Bulk density (g/cm³) |
|---|---|---|---|
| Marsh locations: | | | |
| 2% marsh loss site | 58.1±2.6 | 86.4±0.3 | 0.14±0.01 |
| 11 % marsh loss site | 66.6±1.9 | 85.0±1.0 | 0.17±0.01 |
| 33 % marsh loss site | 51.4±4.2 | 83.3±1.4 | 0.17±0.02 |
| 58 % marsh loss site | 49.0±8.5 | 83.5±2.4 | 0.17±0.03 |
| Lower elevation site | 48.5±3.6 | 84.1±2.2 | 0.16±0.02 |
| Bare patches site | 43.5±4.3 | NA | NA |
| Pond locations: | | | |
| <10 m, unconnected ponds | 43.9±9.7 | NA | NA |
| 10-20 m, unconnected ponds | 44.4±9.8 | NA | NA |
| >20 m, unconnected ponds | 42.3±9.2 | NA | NA |
| >20 m, connected ponds | 39.8±8.5 | NA | NA |

## 3.2 Factors influencing marsh soil shear resistance

Soil shear stength significantly ($p<0.05$) differed between the different field sites, with the highest values found in the 11 % marsh loss site and decreasing towards higher rates of marsh loss (Fig. 3a). The 2 % marsh loss site had a lower soil shear strength than the 11 % marsh loss site. With a linear mixed model, the effect of organic matter content, bulk density, water content, hydroperiod and belowground biomass on shear strength was analysed. From this, only belowground biomass had a significant influence ($p<0.05$), so an additional Pearson correlation test was performed. The belowground biomass and shear vane soil strength of the marsh topsoil samples were strongly correlated (Pearson's correlation $r = 0.91$, $p <0.05$, Fig. 3b). Additionally, we investigated whether the different root fractions had an influence on soil shear strength, but the results indicated that the total root biomass rather than the biomass of individual root fractions were related to soil shear strength.

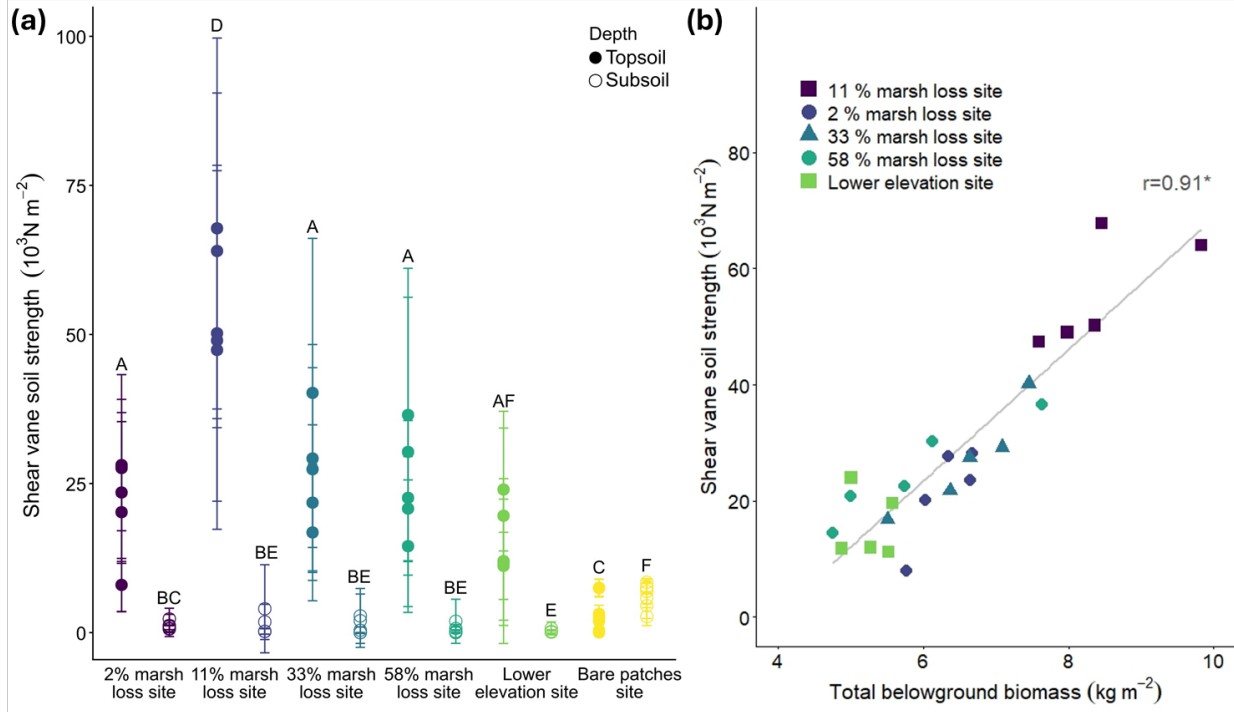

**Figure 3: a) Comparison of shear soil strength ($10^3$ N m$^{-2}$) in the different field locations and between the topsoil (0-10 cm, full circle) and subsoil (30-40 cm, open circle). Letters at the top show the results of the pairwise Wilcoxon rank sum test with Bonferroni correction (n=25 for each site and depth combination), with different letters indicating significant differences between sites and depths. Error bars indicate the standard deviation of each measurement (n=5 for every point) b) Total belowground biomass (kg m$^{-2}$) versus shear vane soil strength ($10^3$ N m$^{-2}$, for 0-10 cm soil depth) for all vegetated marsh sampling locations (no bare or pond locations), demonstrating a strong correlation (r = 0.91, p<0.05).**

### 3.3 Decreasing soil strength with depth

At the marsh sampling locations, we used the penetrologger to examine vertical variation in soil strength in the upper 80 cm of the soil profile. We found that soil strength was maximal between 0-15 cm soil depth and strongly decreased from around 15 cm to 30 cm depth. Below 30 cm the lowest soil strength values were recorded. Across the marsh sites, soil strength (cone penetration resistance) in the top 15 cm of the soil profile (Fig. 4) as well as shear vane soil strength (Fig. 3a) was quite variable. At soil depths below 30 cm this variability between sites was not systematically present anymore (Fig. 4). The shear vane soil strengths at 30 cm depth (<3000 N/m², Fig. 3a) were all consistently lower than the surface measurements (>8000 N/m², Fig. 3a), and there were only very small changes in soil strength at 30 cm depth along the marsh loss gradient (Fig. 3a).

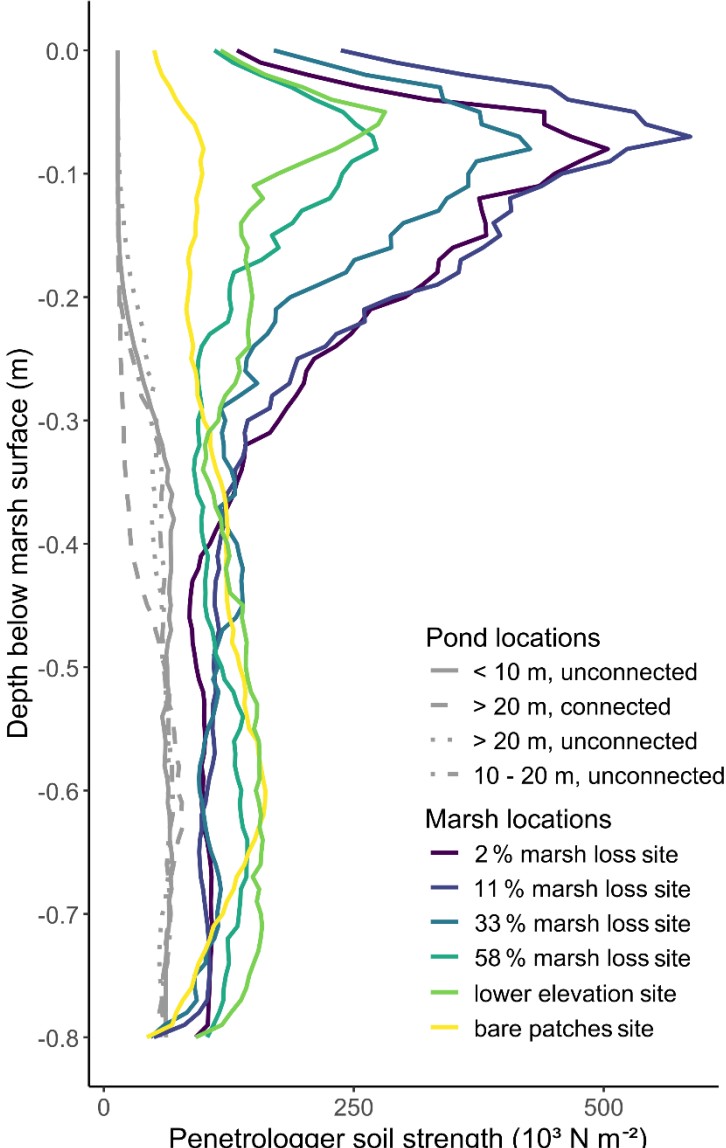

**Figure 4: Penetrologger soil strength ($10^3$ N m$^{-2}$) versus depth below the soil surface (m) for all sampling locations. Soil strength decreases with depth for the vegetated marsh sites. Bare patches and ponds have lower penetrologger soil strength than the marshes at the surface. The y-axis is soil depth relative to marsh soil surface to compare the marsh sampling locations of the different field sites.**

### 3.4 Ponds have low soils strength

The pond topsoils had a much lower soil shear strength (generally below 3000 N m$^{-2}$, Fig. 5) than the vegetated marsh topsoils (8000 to 67000 N m$^{-2}$, Fig. 3a). All the ponds consisted of a loose ooze layer at the top of the soil profile, overlying deeper organic rich layers with a low soil penetration resistance (Figure 4). There does seem to be a difference between the smaller

pond (<20 m) and the larger ponds (>20 m)., where the larger ponds have higher shear strength No rhizomes or stems were found in the pond soil cores, although organic content was high (Table 2, Fig. A1A). Soil organic matter in the ponds consisted of fine microscopic particles compared to the fibrous macroremains of roots, rhizomes and stems of the marsh soil samples (Fig. A1).

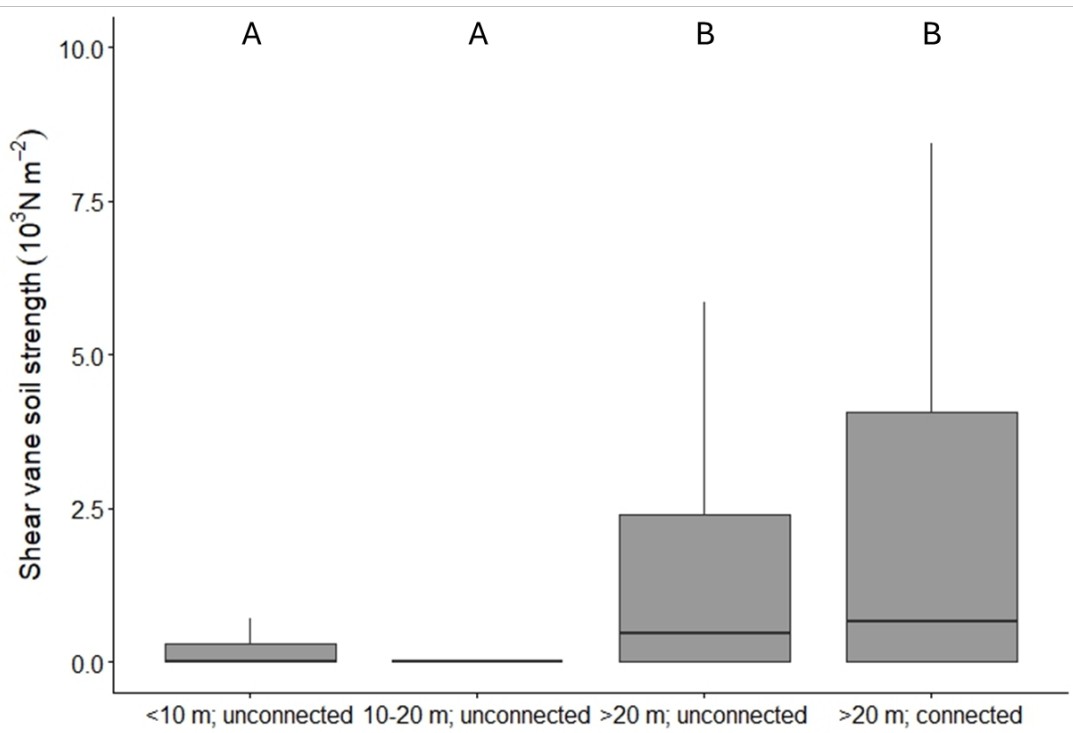

**Figure 5: Shear vane soil strength ($10^3$ N m$^{-2}$) measurements of pond topsoils (n=50 for each boxplot) Significant differences between pond types have different letters above each boxplot, differences between groups have different letters at the very top of the figure (pairwise Wilcoxon rank sum test with Bonferroni correction, $\alpha$= 0.05).**

**4 Discussion**

Coastal marsh conversion into ponds, which may be triggered by sea-level rise, is an important driver of marsh loss. Previous studies on pond expansion within marshes have mainly focused on feedbacks between pond size and wind waves generated on the ponds, as the driving mechanism controlling wave-induced lateral erosion rates of marsh edges surrounding the ponds (Mariotti, 2016, 2020; Ortiz et al., 2017). In this study, we show evidence for an additional potential feedback between sea-level rise, increasing marsh inundation, and decreasing marsh soil strength (measured as shear strength and penetration

resistance), as a potential factor influencing marsh erosion rates. Our field study in a microtidal marsh (with mean tidal range of 0.06-0.63 m) with organic-rich soils (40-70 % organic matter) indicates that (1) an increase in tidal inundation of the marsh

surface (i.e., for a hydroperiod increase from 50 to 95 %) is associated with a loss of soil strength (i.e. decrease in shear strength from around 60 to <10 x $10^3$ N m$^{-2}$ and soil penetration resistance from 450 to <100 $10^3$N m$^{-2}$) of the top soil horizon (0-0.10 m deep) (Fig. 2b); (2) this decrease of the top soil strength is strongly related to the amount of belowground vegetation biomass (Fig. 3b), which is also found to decrease with increasing tidal inundation (i.e. hydroperiod; Fig. 2a); (3) below the soil rooting zone (i.e. upper ca. 0.3 m of the soil profile), a very loose subsoil with weak strength exists (Fig. 3a, Fig. 4); and (4) ponds also have very low top soil strength (Fig. 5). Our finding of decreasing marsh soil strength along a spatial gradient of increasing marsh hydroperiod coincides with a spatial gradient of increasing historical marsh to pond conversion (see Schepers et al. 2017), suggesting that feedbacks between sea-level rise, increasing marsh inundation and decreasing marsh soil strength, may amplify marsh erosion and pond expansion.

Our study is to our knowledge the first providing direct empirical evidence of the relationships between increasing tidal inundation (induced by sea-level rise), decreasing soil strength, and increasing marsh to pond conversion. While we do acknowledge that the observational nature of the study complicates a generalisation of the causal relationships we found, this does not take away that the patterns that we observe are there. Moreover, our findings are confirmed by similar studies, based on marsh locations along a gradient from low to high marsh (Jafari et al., 2024; Stoorvogel, de Smit, et al., 2025; Stoorvogel et al., 2024). For instance, Jafari et al. (2024) and Stoorvogel et al. (2024; 2025) found a decrease in marsh soil strength with increasing tidal hydroperiod along a field gradient from low to high marsh locations. Additionally, combining results from different previous studies indirectly suggests that our finding is qualitatively consistent with previous results.

Our first main finding is the increase in marsh shear strength (Fig. 3b) and penetration resistance (see Appendix, Fig A2) with increasing belowground vegetation biomass. This can be explained by the methodological choice of using a shear vane for soil strength measurements, since roots can be expected to directly affect the shear vane measurements (Brooks et al., 2023). Additionally, since we found a similar relationship between the penetration resistance and belowground biomass, we believe that there is a causal relation. Moreover, this finding generally corresponds with other studies demonstrating that belowground biomass stabilizes the soil against erosion in tidal marshes (Chen et al., 2012; Francalanci et al., 2013; Sasser et al., 2018; Wang et al., 2017) and that vegetated marshes are generally found to experience lower rates of erosion as compared to adjacent bare intertidal sediment surfaces (Gedan et al., 2011; Möller, 2006; Möller et al., 2014; Schoutens et al., 2019).

Secondly, a decrease of above- and belowground biomass production with increasing tidal inundation, above a certain inundation threshold, has been found in several field mesocosm experiments (Kirwan & Guntenspergen, 2015; Langley et al., 2013; Nyman et al., 1994; Voss et al., 2013; Watson et al., 2014), including experiments in our specific study area (Kirwan & Guntenspergen, 2015). Here this inundation-biomass relationship, previously shown by transplantation experiments, is confirmed under undisturbed field conditions for belowground biomass along a spatial gradient of marsh inundation. Furthermore, we also link this inundation-biomass relation to a decrease in soil strength with increasing inundation. Our results

also indicate that the 2 % marsh loss site, with the lowest hydroperiod (on average 29 %), has a lower belowground biomass and lower soil strength than the 11 % marsh loss site with a higher hydroperiod (on average 54 %). For all other field sites with a hydroperiod above 54 %, belowground biomass and soil shear strength are found to decrease with increasing inundation (Fig. 2). This pattern corresponds with the optimum hydroperiod of 35-45 % for which *Schoenoplectus americanus* productivity is found to be maximal in our study area, based on the previous field mesocosm experiments of Kirwan and Guntenspergen (2015). *S. americanus* is considered a low marsh species (Broome et al., 1995; Kirwan & Guntenspergen, 2015; Nyman et al., 1994) and previous research indicates that *S. americanus* productivity is reduced when it grows under a low hydroperiod (Kirwan & Guntenspergen, 2015; Nyman et al., 1994). Kirwan and Guntenspergen (2015) also concluded that the optimal hydroperiod for belowground productivity of *S. americanus* is between 35 and 45 % as determined in an experimental setup close to the 2 % and 58 % marsh loss sites respectively (indicated by the dashed lines in Fig. 2a and b) and that lower or higher hydroperiods lead to lower root productivity. The 2 % marsh loss site does have a hydroperiod below this optimum (<30 %, Fig. 2a and Table 1), whereas all other field sites have a hydroperiod above that optimum (>50%), which may explain why the 2 % marsh loss site has a lower belowground biomass and soil strength as compared to the 11 % marsh loss site, and why a decreasing soil strength with increasing hydroperiod above 50 % is found (Fig. 2).

In our vegetated sampling locations, we found that the roots provide structural soil strength in the upper 15 cm of the soil profile, which is confirmed by multiple other studies (Brooks et al., 2022; Lo et al., 2017). However, below this threshold depth, both root biomass and soil strength (Fig. 3a, Fig. 4) rapidly decrease. Although we took soil samples and determined the biomass of only the upper 15 cm, several other studies in tidal marshes suggest that the majority of the rhizomes and roots are situated in the top 15 cm of the soil profile (Saunders et al., 2006; Valiela et al., 1976). This implies that the vertical distribution of belowground biomass also determines the vertical variation in soil strength. Similar findings on vertical soil strength variation have been reported in our specific study area (Stevenson et al. 1985) and in the North Inlet estuary in South Carolina (Jafari et al., 2024). In other studies, sediment properties, such as organic matter content, bulk density and clay content have been shown to play a role in the cohesion of sediment and thus the shear strength (Feagin et al., 2009; Gillen et al., 2021; Joensuu et al., 2018). Higher organic matter content may increase the sediment erosion resistance, which corresponds to our finding of higher organic matter content in the sites with higher shear and penetration resistance. Studies have shown that both higher bulk density and clay content decrease the erodibility of the marsh sediment (Brooks et al., 2022; Feagin et al., 2009b; Gillen et al., 2021; Lo et al., 2017; Stoorvogel, de Smit, et al., 2025). These studies are however located in minerogenic marsh systems, where bulk densities and clay contents are generally higher than in organogenic systems as ours. Therefor we believe that the influence of belowground biomass on shear and penetration resistance will dominate over the effect of bulk density and clay content in organogenic systems.

The presence of a weak subsoil below the upper root zone, implies that local vegetation disturbances, bare patches or early ponds, may allow exposure of the weak subsoil to erosion. Moreover, once ponds are formed, we may expect that the marsh

edges surrounding the ponds are vulnerable to increased erodibility of the exposed weaker subsoil, which may promote undercutting (i.e. erosion of the subsoil layer) of the rooted top layer and subsequent cantilever failures (i.e. when the topsoil block remaining after undercutting collapses), a mechanism that is found to be important in driving lateral erosion of scarped marsh edges with undercutting (Bendoni et al., 2016). Indeed, the pond edges in our study area have steep scarps (Schepers et al. 2020a), which makes them vulnerable for wave attack and potential undercutting and cantilever failures once the wind fetch

length is large enough.

Our results also indicate that pond bottoms have particularly weak soils. Based on the findings of Stoorvogel, Willemsen, et al. (2025), where both shear strength and erosion were studied, that even relatively small differences in shear strength can correspond with large differences in erosion rates, we assume that our pond bottoms are very vulnerable to erosion. First, the

360 pond bottom material is composed of much more fragmented, organic-rich material that has likely formed through decomposition of the originally vegetated marsh soils after conversion of vegetated marshes into bare patches and ponds (DeLaune et al., 1994; Stevenson et al., 1985; van Huissteden & van de Plassche, 1998). This results in a loose unconsolidated layer with low strength at the bottom of the ponds (Fig. 4 and 5). This seems to be a typical property of interior marsh ponds comparable to findings in salt marshes in Maine (Wilson et al. 2010). We hypothesize that the loose layer may be easily

suspended by waves and tidal currents, and when ponds are connected to the tidal channel system, this might facilitate the tidal transport of the suspended material out of the ponds (Schepers et al. 2020a) and further in seaward or bay-ward direction out of the marsh system, as indicated by sediment flux measurements in the tidal channels in the studied marsh system (Ganju et al., 2013, 2017). As such, the easily eroded material from the pond bottom or below the vegetated root zone may be removed and may enable further deepening of ponds. The deep ponds in our study area are permanently submerged, given the low tidal

range (Table 1), hence preventing pioneer marsh plants from reestablishing and protecting the cliffs against further erosion, a defense that has been observed in other marsh systems (van de Koppel et al., 2005; van der Wal et al., 2008; Wang et al., 2017). These findings indicate that ponds, once they are formed, are prone to erosion and that recovery of marsh vegetation is very unlikely (Schepers et al. 2020a).

Together, these results suggest a potential new feedback for the formation and expansion of small marsh ponds, in which increasing inundation drives weaker marsh soils, which increases erodibility of the marsh, hence promoting formation and enlargement of ponds. Small marsh ponds typically originate near drainage divides at far distances from tidal creeks, where sedimentation rates are low, and vegetation mortality is associated with poorly drained soils (Redfield, 1972; Schepers et al., 2017; Vinent et al., 2021). However, the growth of these small interior ponds is poorly understood because the ponds are

located far from sources of erosion, such as tidal channels and waves. Thus, pond expansion is thought to occur largely through passive drowning and merging of individual small ponds (Himmelstein et al., 2021; Schepers et al., 2017), until ponds are large enough that they intersect the tidal channel network and/or become vulnerable to wave erosion (Mariotti, 2016, 2020; Schepers et al., 2020a). Our work suggests an additional, more dynamic response, where inundation leads to more erodible

sediment. Proposed feedbacks linking pond growth to wind fetch-driven erosion are most applicable to very large ponds that
exceed a critical length for the formation of wind waves (i.e. >200 m – 1 km in length) (Mariotti & Fagherazzi, 2013; Ortiz et al., 2017). Yet, elongation of ponds in directions of dominant wind occur for smaller pond sizes in our study area (i.e. ponds of about 100 x 100 m in size) (Stevenson et al., 1985). Thus, our finding that shear strength decreases with increasing inundation suggests that critical wind fetch lengths for runaway erosion may be smaller than otherwise anticipated and offer a potential explanation for the growth of much smaller ponds.

## 4.1 Limitations of the study

A first limitation of our study is the use of a space-for-time substitution, assuming that a spatial gradient in increasing marsh inundation and increasing pond area can be considered representative for the temporal development of increasing pond surface within a marsh, as a result of increasing marsh inundation in response to sea level rise. Because of this space-for-time approach,
there could be differences between sites, other than differences in inundation and pond surface area, that could influence the vegetation belowground biomass production that we have not considered. However, given the qualitative agreement of our results with previous findings who don't use this space-for-time substitution (as discussed above), we believe that this effect is limited.

Further, the use of shear vane devices is not recommended for direct comparison between different studies, as measurements are influenced by the present roots, but also the person who takes the measurements. We therefore recommend on the one hand that shear vane devices are used in combination with other methods for evaluating soil strength, such as a penetrologger (used in our study) or a Cohesive Strength Meter (Brooks et al., 2023). On the other hand, we recommend to only compare patterns and not absolute values between studies. We argue however that when measurements are performed by the same person, shear
vane measurements are valid for comparison of relative differences in sediment bed strength within a given study area, as done in our study.

Finally, we recognise that other environmental variables, which are not considered in our study, could influence vertical variations in soil strength. For instance, higher water content has been shown to decrease the soil penetration resistance (Gillen
et al., 2021; Stoorvogel, de Smit, et al., 2025). As soil water content may be higher in deeper soil layers, this may also contribute to lower soil strength deeper in the profile. Yet, we expect this plays a minor role in our study sites as field observations typically indicate water saturated soils over the whole soil profile. Additionally, variations in soil strength along the spatial marsh degradation gradient may be related to factors we did not account for. For instance, higher nutrient loading decreases the soil organic matter content and belowground vegetation biomass and has been reported to be related to reduced soil strength
(Turner et al., 2020). Bioturbation, especially burrowing by crabs, can increase the oxygenation of the sediment and facilitate the breakdown of belowground biomass (Wilson et al., 2012). Yet we have no data to test whether such factors varied along

the spatial marsh degradation gradient and if they contributed to the observed spatial pattern of decreasing soil strength with increasing marsh degradation.

**5 Conclusion**

Our study demonstrates that excessive tidal inundation above a threshold (here above a hydroperiod of about 50 %) leads to weaker soils in a microtidal, organic-rich marsh system. We found that the soil strength is strongly related to the amount of belowground biomass, especially the macroscopic fraction consisting of roots, rhizomes and stem fragments, which consists of fibrous interconnected material that provides structural stability to marsh soils. Moreover, below the shallow rooting zone and at the bottom of interior marsh ponds the soil is not cohesive and very weak, which may amplify expansion and deepening

of ponds, and may contribute to further marsh loss. Our finding of decreasing marsh soil strength along a spatial gradient of increasing marsh inundation coincides with a gradient of increasing historical marsh loss by pond expansion, suggesting that feedbacks between sea-level rise, increasing marsh inundation and decreasing marsh soil strength, may amplify marsh erosion and pond expansion.

**Acknowledgements**

This project was financed by an UA-BOF DOCPRO grant (to L.S. and S.T.), the Research Foundation Flanders (FWO, PhD grants L.S., 11S9614N & 11S9616N, travel grants L.S. K220916N, project grant nrs. G060018N & G031620N), by the U.S. Geological Survey, Ecosystems Land Change Science Program (G.G.), by NSF GLD 1529245, NSF SEES 1426981, NSF LTER 1237733 (M.K.). We would like to thank the managers and biologists of the Blackwater National Wildlife Refuge for their assistance and valuable comments; Patrick Brennand and Liza McFarland (USGS) for indispensable field assistance.

Steven Bouillon (KULeuven) assisted in analyzing soil samples. Any use of trade, firm, or product names is for descriptive purposes only and does not imply endorsement by the U.S. Government.

**Author contribution statement**

LS and ST conceptualised the study, with the help of MK and GG. LS carried out the fieldwork and lab analysis, with resources provided by MK and GG. MH and LS analysed and visualised the data. LS, ST and MH prepared the manuscript with

contributions from all the co-authors.

**Data availability**

All data is available in this repository:

https://zenodo.org/records/17206424?token=eyJhbGciOiJIUzUxMiJ9.eyJpZCI6ImQxOWI5YzE1LTQ5NDAtNDRmMC04MDMxLTQxZWRhNTMxZDAzNSIsImRhdGEiOnt9LCJyYW5kb20iOiJmMGY3NjJjYzJmNjM0ZTVlM2QwOGM5NDEzY2Y2MmE1MCJ9._6pVHBB6ACU5qjJ2_w574pNfFc2NEpcNqOUaZMGs2hnmLDumMOX5LP4GqBAvuARmGvCw_5m_ldtTzOdOQUrs6A

## Competing interests

The authors declare that they have no conflict of interest.

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

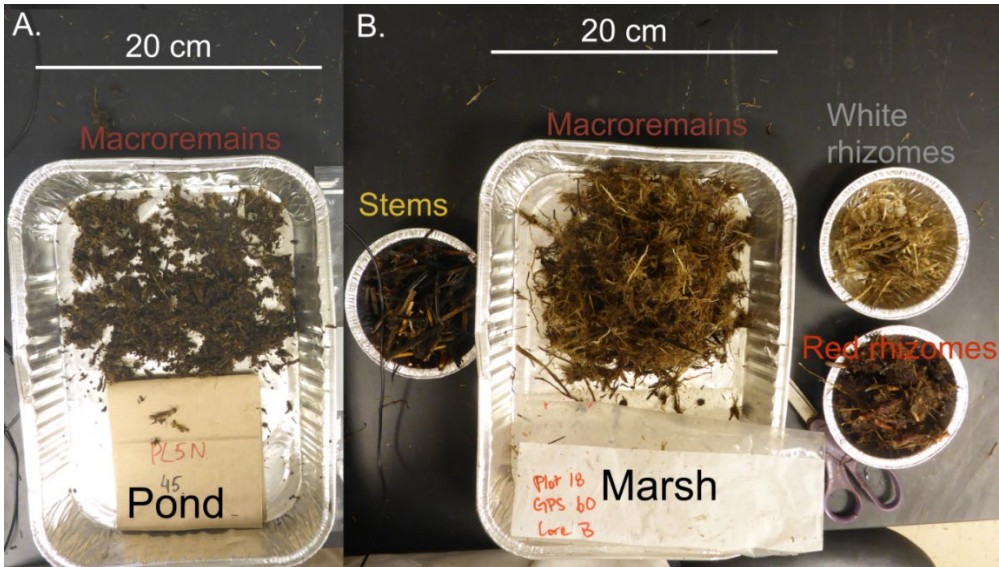

**Figure A1: Biomass retrieved from a pond core (A) and a marsh core (B). Only macroremains (neither rhizome nor stem, but >2 mm) were present in the pond sample (A), which were much more fragmented compared to the fibrous macroremains of the marsh sample (B).**

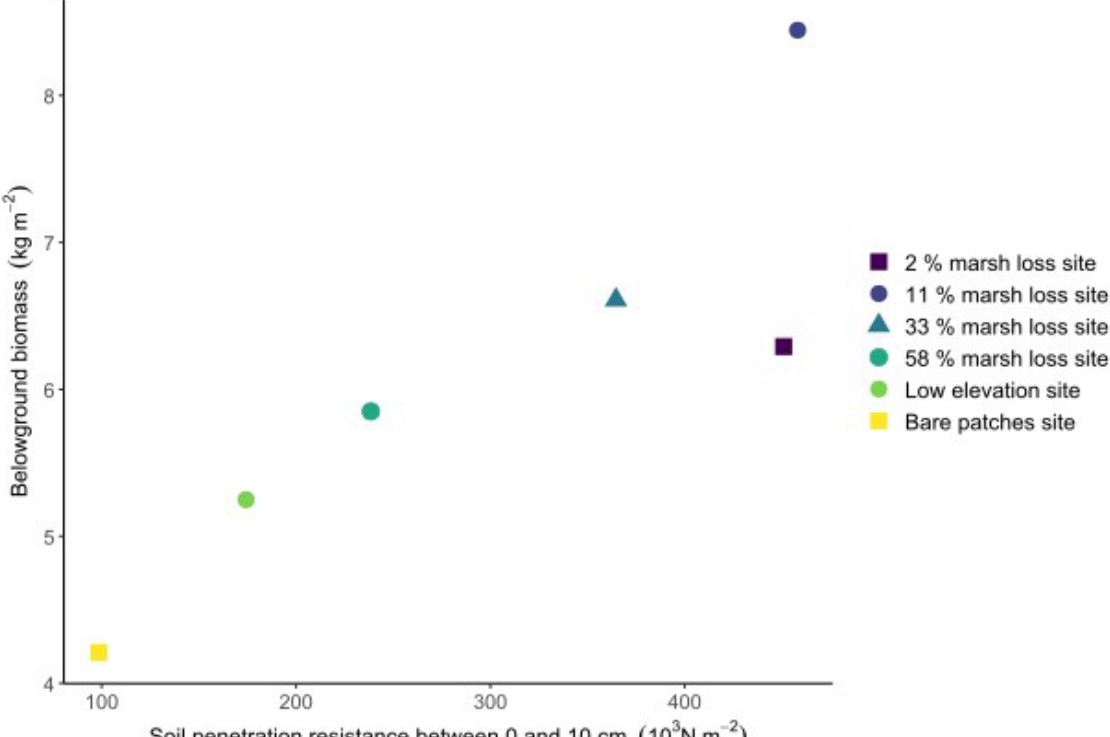

**Figure A2: Relation between soil penetration resistance ($10^3$N m$^{-2}$) in the top soil and the belowground biomass (kg m$^{-2}$) for the different field sites.**