# Peer review of "Sea-level rise in a coastal marsh: linking increasing tidal inundation, decreasing soil strength and increasing pond expansion"

_EGUsphere, 2025_

## Author Comment (AC1)

**Reviewer 1**

Schepers et al. look to link tidal inundation from sea-level rise to decreasing soil strength in salt marshes, which can be related to the loss of belowground biomass from the increasing tidal inundation. The findings of this paper make sense based on what we know from previous work; however, the methods used here have issues/limitations that need to be addressed. There are also other variables that could impact soil strength in these marshes that are not discussed. Additionally, I believe the authors undervalue previous marsh work in the investigation of tidal inundation on soil strength.

*We would like to thank the reviewer for the very insightful and critical comments on our manuscript. We have tried to include them in the manuscript.*

Major Comments:

- Much of the paper's discussion and findings is based on the use of a shear vane device to interpret soil stability. This includes findings that belowground biomass increases soil strength. However, I think the authors need to acknowledge the issues related to using a shear vane as an indicator of soil stability, including that roots within the soils will directly impact the vane measurements. Hence it would make sense that there is a relationship between belowground biomass and soil/root strength if roots impact these measurements. Vane measurements can also be impacted by how fast the vane is rotated, which can vary across users.

  *Thank you for this valuable comment. We have added a part in the discussion highlighting that a relation between shear vane shear strength and belowground biomass may be expected due to the methodology, but we also highlight that the penetrologger measurements also indicated an increasing soil penetration resistance with increasing biomass:*

  Lines 316-321: "Our first main finding is the increase in marsh shear strength (Fig. 3b) and penetration resistance (see Appendix, Fig A2) with increasing belowground vegetation biomass. This can be partly explained by the methodological choice of using a shear vane for soil strength measurements, since roots can be expected to directly affect the shear vane measurements (Brooks et al., 2023). Additionally, since we found a similar relationship between the penetration resistance and belowground biomass, we believe that there is a causal relation."

  *Additionally, as we were aware of the differences that can occur between different people taking the measurements, it was assured that they were taken by the same person each time while conducting the fieldwork.*

- The paper also is missing context as to what the differences in soil strength really mean between sites. From a soil strength perspective, is the difference in strengths depicted in Figure 3 and Figure 5 really large enough to generate significant differences in the erodibility of the soil? And how do these differences compare to other studies? Comparing Figure 3 to the penetrologger data in Figure 4, the penetrologger differences are much larger than the vane differences so how does this fit into the narrative? The penetrologger data is not used in this paper as much as the vane data.

*Thank you for this comment. We have split this answer into different points to respond to each question separately.*

*From a soil strength perspective, is the difference in strengths depicted in Figure 3 and Figure 5 really large enough to generate significant differences in the erodibility of the soil?*

- *Since the shear strength ranges from >60 to less than $10 . 10^3 N/m^2$ and other salt marsh studies hardly ever report values above 60, we do believe that this shows a significant decrease in shear strength. Additionally, if you go to these field sites, the difference in soil stability are immediately felt. The sites with the highest shear strength values are very easy to walk on (our boots are not much sinking into the soil ), while the lowest shear strength sites have very soft soils (if you take a wrong step you can sink about 50 cm into the sediment).*

*How do these differences compare to other studies?*

- *According to a recent paper by Brooks et al. 2023 the direct comparison of shear strength values between studies is difficult, since often different methods are used that can significantly influence the absolute values. Additionally, as you mentioned in a previous comment, the shear vane measurements are impacted by the person performing the measurements. Therefor we have chosen not to directly compare our values with other values, but focussing on the relative differences we see within our study. However we did check whether our shear vane values were in the same range as other publications, which is the case.*

- Comparing Figure 3 to the penetrologger data in Figure 4, the penetrologger differences are much larger than the vane differences so how does this fit into the narrative? The penetrologger data is not used in this paper as much as the vane data.

  - *The penentrologger measured the resistance of the soil to penetration, while the shear vane measured the shear resistance of the soil. These are two different measures of sediment strength and two different measurement methods, so differences between them are logical. However, the trends we see in the shear vane data, like the decrease in strength with depth, are also observed in the penetrologger data. Further, both the shear vane and penetrologger data show an increasing soil strength with increasing below-ground biomass. We have integrated the penetrologger data more in the discussion.*

    Line 295-297: "Our field study in a microtidal marsh (with mean tidal range of 0.06-0.63 m) with organic-rich soils (40-70 % organic matter) indicates that (1) an increase in tidal inundation of the marsh surface (i.e., for a hydroperiod increase from 50 to 95 %) is associated with a loss of soil strength (i.e. decrease in shear strength from around 60 to $<10 \times 10^3$ N m$^{-2}$ and soil penetration resistance from 450 to $<100$ $10^3$N m$^{-2}$) of the top soil horizon (0-0.10 m deep) (Fig. 2b);…"

    Line 319-321: "Additionally, since we found a similar relationship between the penetration resistance and belowground biomass, we believe that there is a causal relation."

- For the penetrologger profiles in Figure 4, which have the max strength near the surface, are there other factors other than roots that can also contribute to these differences along depth? For example, how does soil moisture change downcore? You only provide soil

moisture data for the topsoil. Where is the BD and OM data? Also, although you may no longer have live roots at deeper depths, you would expect to see dead roots along the profile that should also impact soil strength.

*Thank you for this very accurate question. Water content, bulk density and organic matter were only analysed in the top 15 cm. However, data from a more recent study in the Blackwater marshes (Huyzentruyt et al., in review) show that the bulk density is fairly constant over a depth of 40 cm. There is however some variation of organic carbon with depth, but without a clear trend along the gradient of increasing hydroperiod (sometimes there is a slight increase with depth, other times a decrease). We may not exclude the presence of dead roots at depth, which could indeed influence the strength, but we do believe that this effect on the penetration resistance is lower than for living roots (giving the lack of turgor pressure for example). We have done some additional testing with the OM and BD data and added some additional discussion on these points (see our reply to comment below).*

Line 189-196: "2.5 Statistical analysis

The effect of hydroperiod on shear strength and belowground biomass was analysed using linear mixed models (LMM), using field site as a random effect to account for within site clustering. A separate LMM analysis was performed to evaluate the influence of organic matter content, bulk density, water content, hydroperiod and belowground biomass on shear strength, again incorporating field site as a random effect. The differences in bulk density, water content, organic matter, shear strength and belowground biomass between sites were analysed using pairwise Wilcoxon rank sum test with Bonferroni correction. All analyses were executed in R (R core team, 2022), using the lme4 package (Bates et al., 2015) for the linear mixed models. The p-value threshold used is 0.05. "

- There are other variables not mentioned by the authors that could impact soil strength rather than just belowground biomass, including potential differences in bioturbation and differences in grain size distributions. There are also other factors other than tidal inundation that impact belowground biomass, including nutrient loadings, which can vary even along a marsh site.

*Thank you for this valuable comment. We have added a paragraph to the discussion integrating more potential influences on soil strength:*

Line 362-379: "We recognise that other factors, which are not considered in our study, may influence vertical variations in soil strength. For instance, higher water content has been shown to decrease the soil penetration resistance (Gillen et al., 2021; Stoorvogel et al., 2025). As soil water content may be higher in deeper soil layers, this may also contribute to lower soil strength deeper in the profile. Yet we expect this plays a minor role as field observations typically indicate water saturated soils over the whole soil profile. Additionally, variations in soil strength along the spatial marsh degradation gradient may be related to factors we did not account for. For instance, higher nutrient loading has been shown to decrease the soil organic matter content and belowground vegetation biomass and has been reported to be related to reduced soil strength (Turner et al., 2020). Bioturbation, especially burrowing by crabs, can increase the oxygenation of the sediment and facilitate the breakdown of belowground biomass (Wilson et al., 2012). Yet we have no data to test whether such factors varied along the spatial marsh degradation gradient and if they contributed to the observed spatial pattern of decreasing soil strength with

increasing marsh degradation. Lastly, sediment properties such as organic matter content, bulk density and clay content may play a role in the cohesion of sediment (Feagin et al., 2009; Gillen et al., 2021; Joensuu et al., 2018). Higher organic matter content may increase the sediment erosion resistance, which corresponds to our finding of higher organic matter content in the sites with higher shear and penetration resistance. Studies have shown that both higher bulk density and clay content decrease the erodibility of the marsh sediment (Brooks et al., 2022; Feagin et al., 2009b; Gillen et al., 2021; Lo et al., 2017; Stoorvogel et al., 2025). These studies are however located in minerogenic marsh systems, where bulk densities and clay contents are generally higher than in organogenic systems as ours. Therefor we believe that the influence of belowground biomass on shear and penetration resistance will dominate over the effect of bulk density and clay content."

- Data needs to be made publicly available either through supplemental materials or a data repository before this paper can be published.

  *You are correct. We are working on compiling the data, so that by the time of publication a public repository will exist with the data.*

Minor Comments:

- Sea-level rise should be hyphenated throughout.

  *We have adjusted the text so that sea-level rise is always hyphenated*

- Figure 1 can be improved by 1) making the symbol size larger (the symbols in 1b are too small); 2) making some of the text larger, including the labels of a, b, and c in the Figures; 3) adding in a symbol to indicate direction (e.g. a north arrow); 4) correcting the a,b,c labels in the figure caption to be consistent with the labeling in the figures. Also check with the journal requirements to see what type of labeling is required.

  *Thank you for the suggestions. The changes have been made to figure 1 and adjusted in the text.*

  Line 86-87: The mean tidal range decreases from 63 cm at Fishing Bay (bottom right of Fig. 1a )  to 6 cm at Lake Blackwater (top left of Fig. 1a )

  Line 92-98: **Figure 1: a: Aerial images of the Blackwater marshes (black: water, light grey: marsh) with sampling locations (Copernicus – Sentinel data [2025]. Retrieved from Google Earth Engine, processed by ESA). The marsh loss (i.e. proportion of shallow open water ponds to total marsh area) is quantified for each site based on Schepers et al. (2017) . The inset map shows the location of the Blackwater marshes in the Chesapeake Bay. The green box is the extent of panel  b: pond locations (white) sampled at site 4. Values in the legend of (b) refer to the average pond diameter in each category. The yellow box is the extent of panel  c. c: marsh locations at the 58 % marsh loss site  with (green) and without (yellow) vegetation.**

- Line 107 – yr-1 should be made a superscript and applied throughout the rest of the manuscript.

  *Thank you for noticing, it has now been changed accordingly*

- Also line 107 – Has the rate of SLR changed over time?

  *Thank you for this question. No there is no evidence for a change in rate of SLR since the 1940s. The data (below) do not show an increase or decrease in the rate of SLR over this*

*period but a rather linear trend (source: NOAA tides and currents data). This was included in the text:*

Line 113: "are less than the relatively constant long-term rate of relative sea-level rise of 4.06 mm yr$^{-1}$ in Cambridge, MD, 1943-2025..."

[Figure]

- Section 2.3. – Later you bring up measurements down to 80 cm but you don't mention this anywhere in this section.

  *Thank you for noticing, we have added a more detailed description in section 2.3.*

  Line 164-166: "The measurement was taken at all marsh sites (five plots in the 2 %, 11 % and 33 % marsh loss site and 17 in the 58 % marsh loss site) an pond sites in the upper 80 cm of sediment."

- Section 2.4. heading should be changed to include soil C

  *Thank you for the suggestion, we have changed the title of Section 2.4 to:*

  Line 168: "Belowground biomass sampling and sediment analysis"

- Why only collect soils to 15cm depth when you measure soil strength to 30cm?

  *Thank you for this question, as this is indeed not clear from the text. The shear vane soil strength was measured at the top of the profile (0-10 cm) to investigate the relationship between soil shear strength and the amount of belowground biomass. The 30 cm measurement was assumed to be below the rooting depth of Schoenoplectus americanus, as to try and see the effect of belowground biomass on the soil shear strength. This assumption has also been verified in a field campaign for a different study in this area, where we noticed living roots were hardly present anymore at deeper depths.*

- How much time was soil ashed at 550 to determine organic content?

  *We have added more specifications on the LOI protocol in the methods as follows:*

  Line 177: "One half was dried, ground and homogenized with a 0.5-mm grinder (Retsch ZM2000) and heated to 550°C and ashed for four hours to determine the organic content of the soil samples (loss on ignition)."

- Organic content is barely mentioned in the manuscript

*Thank you for the valuable comment. We have extended the analysis on organic matter by looking at differences between sites and including it in the linear mixed model for shear strength. In correspondence with comments from you and reviewer 2, we have added a paragraph describing other potential influencing factors, among which organic matter. (see comment 2).*

- Table 1 – is the font in the last line of the caption different from the rest of the caption? It looks smaller to me.

*Thank you for noticing, we have corrected the font size so that it matches.*

- Table 2 – font size of this table looks different from the other tables.

*Thank you for noticing, we have corrected the font size so that it matches.*

- Figure 3 – the gray and yellow coefficients are hard to read and all coefficients can be made larger.

*Thank you for the suggestion. In accordance with the comments of reviewer 2, we have changed the content of fig 3 (fig 3A became fig 3b and 3b was omitted). Additionally, we increased the size and darkened the color of the coefficient in the previous figure 3a according to your comment.*

- Line 223 – unclear what difference the authors are referring to.

*Thank you for the valuable suggestion. We have changed the phrase as follows:*

Line 264: "At soil depths below 30 cm this  variability between sites was not systematically present anymore (Fig. 4)."

- Figure 5 – Why are you comparing pond soils and bare patches to just the 30cm marsh depth rather than the 15cm marsh depth?

*Thank you for this question. Since pond bottoms and bare patches don't contain living vegetation, we can also assume that there are hardly any living belowground biomass left. Therefor it seems more logical for the figure to compare these values with the 30 cm depth values, as they were taken to represent densities below the rooting depth of Schoenoplectus americanus. In the text however, we do also compare the results to the top 15 cm marsh results shown in figure 3.*

- Line 260 – Do studies that examine soil strength with tidal inundation along a marsh elevation gradient not count?

*Thank you for the question. We have emphasised the difference between our and other studies a bit more:*

Line 306-310: "Our study is to our knowledge the first providing direct empirical evidence of the relationships between increasing tidal inundation (induced by sea level rise), decreasing soil strength, and increasing marsh to pond conversion. Our study is observational and conducted in a specific system (a micro-tidal marsh with organic-rich soils), which intrinsically limits drawing generalized conclusions and causal relationships. While we do acknowledge such limitations, this does not take away that the relationships that we observe are there. Moreover, our findings are in line with other studies,

 based on a comparison of soil strength between marsh locations along a gradient from low to high marsh. For instance, Jafari et al. (2024) and Stoorvogel et al. (2024; 2025) found a decrease in marsh soil strength with increasing tidal hydroperiod along a field gradient from low to high marsh locations."

- Line 261 – seems like a word or two is missing here.

  *Thank you for noticing. Based on the previous comment and a comment of reviewer 2 this sentence is now changed and the issue of the missing words is resolved.*

---

## Author Comment (AC2)

**Reviewer 2**

In their manuscript *Sea level rise in a coastal marsh: linking increasing tidal inundation, decreasing soil strength and increasing pond expansion*, Schepers et al. highlight an interesting mechanism – previously demonstrated experimentally – by which sea level rise and associated increases in tidal inundation may promote pond expansion and reduce vegetated marsh area via declines in soil strength, likely driven by reduced belowground biomass. I found the manuscript engaging and generally well written. However, I have several major and minor comments that I strongly encourage the authors to address. In particular, the observational nature of the study – and its inherent limitations – along with the analytical approach, require clearer justification and discussion.

*We would like to thank the reviewer for the very insightful and critical comments on our manuscript. We have tried to include them in the manuscript.*

**MAJOR COMMENTS**

1) The study is observational and conducted in a specific system (microtidal marsh with organic-rich soils), which limits causal inference. This is important to emphasize when comparing with other marsh types and when interpreting findings. Please add a paragraph in the Discussion outlining the study's limitations and potentially suggesting next steps. For instance, in L121-122, while site-specific elevation is acceptable, elevation-driven variation may influence soil strength. Unlike experimental approaches that isolate variables, your study interprets a natural gradient with inherent co-variation. This is valuable, but should be framed as such.

*Thank you for the valuable comment. We have integrated your valid point into our discussion to highlight the observational nature and site-specific nature of our study and associated limitations:*

Line 306-310: "Our study is to our knowledge the first providing direct empirical evidence of the relationships between increasing tidal inundation (induced by sea-level rise) , decreasing soil strength, and increasing marsh to pond conversion. Our study is observational and conducted in a specific system (a micro-tidal marsh with organic-rich soils), which intrinsically limits drawing generalized conclusions and causal relationships. While we do acknowledge such limitations, this does not take away that the relationships that we observe are there. Moreover, our findings are in line with other studies,  based on marsh locations along a gradient from low to high marsh. For instance, Jafari et al. (2024) and Stoorvogel et al. (2024; 2025) found a decrease in marsh soil strength with increasing tidal hydroperiod along a field gradient from low to high marsh locations."

*Further we have added the limitations you mentioned in the text where they apply.*

On the use of shear vane measurements. Line 316-319: "Our first main finding is the increase in marsh shear strength (Fig. 3b) and penetration resistance (see Appendix, Fig A2) with increasing belowground vegetation biomass. This can be partly explained by the methodological choice of using a shear vane for soil strength measurements, since roots can be expected to directly affect the shear vane measurements (Brooks et al., 2023)."

On the space-for-time substitution. Line 350-352: "Of course, since we are using a space-for-time substitution, there could be other differences between sites (such as salinity and tidal range) that could influence the vegetation belowground biomass production, however given the agreement of our results with these previous findings, we believe that this effect is limited."

2) Your replication is at the site level, but the 5 locations within sites are treated as independent replicates. Thus, your statistical inference (but see Comments 3 and 4) is confounded with site. While I recognize the logistical constraints of field ecology, please acknowledge this in your limitations section (see Comment 1) and clarify the implications for interpreting your results.

*Thank you for the valuable advise. We have changed our statistical analysis to linear mixed models, to include the random effect of site. See also response to comment 3 and 4.*

3) There is no dedicated paragraph describing your statistical analysis. Readers need a clear overview of how hypotheses were tested, which variables were used, whether data were averaged, and which models were applied. Please add a paragraph detailing your statistical approach.

*Thank you for the very valid suggestion. We have added a paragraph on the statistics:*

Line 189-196: "2.5 Statistical analysis

The effect of hydroperiod on shear strength and belowground biomass was analysed using linear mixed models (LMM), using field site as a random effect to account for within site clustering. A separate LMM analysis was performed to evaluate the influence of organic matter content, bulk density, water content, hydroperiod and belowground biomass on shear strength, again incorporating field site as a random effect. The differences in bulk density, water content, organic matter, shear strength and belowground biomass between sites were analysed using pairwise Wilcoxon rank sum test with Bonferroni correction. All analyses were executed in R (R core team, 2022), using the lme4 package (Bates et al., 2015) for the linear mixed models. The p-value threshold used is 0.05. "

4) Related to the above: while correlation may be suitable for Fig. 3, hydroperiod is unlikely to be a response variable influenced by belowground biomass or soil strength. Therefore, regression would be more appropriate to suggest directional relationships in Fig. 2. If you intentionally chose correlation, please explain why. See also Comment 3 regarding the missing statistical analysis section.

*Thank you for the advice. We have changed the statistical analysis from only correlation testing to also include linear mixed models (see response above).*

**RELATIVELY MINOR COMMENTS**

5) L14: Replace "method" with "mechanism"; rephrase the sentence accordingly.

*Thank you for the suggestion. We have changed the word.*

Line 14: "Here, we propose another  mechanism…"

6) L14 and L51: Define "soil strength" clearly at first appearance in both the abstract and main text.

*Thank you for the valuable suggestion. We have added a more detailed explanation on what is meant by "soil strength" both in the abstract and the main text.*

Line 14-15 *"Here, we propose another  mechanism between sea-level rise, increasing marsh inundation, and decreasing marsh soil strength (approximated here as resistance to shear and penetration stress),..."*

Line 55:*"...we investigate the hypothesis that the marsh soil strength (measured as resistance against shear and penetration stress)..."*

7) L69-70: Clarify what is meant by "stable marsh system." Do you mean a system not subject to sea level rise?

*Thank you for the suggestion. We have changed the text as such:*

Line 72: *"This relationship was however quantified in a  marsh system without signs of degradation as a result of sea level rise"*

8) L70: The phrasing suggests a direct link between soil strength, marsh loss, erosion, and pond expansion. Please revise to reflect the uncertainty of these associations.

*Thank you for the suggestion. We have changed the sentence as such:*

Line 78: *"Our analysis  suggests relationships between..."*

9) L72: "Microtidal" should be hyphenated or not consistently throughout. Ensure consistency in terminology across the manuscript.

*Thank you for noticing. The non-hyphenated version has been used throughout the whole text.*

10) L82-84: Rephrase this section to introduce the experimental design first (number of sites, selection criteria) before referring to site numbers.

*Thank you for the valuable suggestion. After all we have removed the mentioning of site numbers:*

Line 86-87: "The mean tidal range decreases from 63 cm at Fishing Bay (bottom right of Fig. 1a) to 6 cm at Lake Blackwater (top left of Fig. 1a)"

11) L88-93 (Figure caption): (1) Capitalization of letters should match figure; (2) use "panel" instead of "figure" for subplots; (3) explain the inset; (4) clarify data source for marsh loss values.

*Thank you for the suggestion. The caption has been changed accordingly.*

Line 92-98: **Figure 1: A: Aerial images of the Blackwater marshes (black: water, light grey: marsh) with sampling locations (Copernicus – Sentinel data [2025]. Retrieved from Google Earth Engine, processed by ESA). The marsh loss (i.e. proportion of shallow open water ponds to total marsh area) is quantified for each site based on Schepers et al. (2017) . The inset map shows the location of the Blackwater marshes in the Chesapeake Bay. The green box is the extent of panel  b. b: pond locations (white) sampled at site 4. Values in the legend of (b) refer to the average pond diameter in each category. The yellow box is the extent of panel  c. C: marsh locations at the 58 % marsh loss site  with (green) and without (yellow) vegetation.**

12) L97-98: If you use cardinal directions, mark them on the figure. Otherwise, use terms like "left" or "right."

*We added a northern arrow to the map to resolve this issue.*

13) L104: Acknowledge limitations of space-for-time substitution briefly here and expand in Discussion (see Comment 1).

*Thank you for this valuable comment. We have added some nuance on the space-for-time substitution in the discussion:*

Line 350-352: "Of course, since we are using a space-for-time substitution, there could be other differences between sites (such as salinity and tidal range) that could influence the vegetation belowground biomass production, however given the agreement of our results with these previous findings, we believe that this effect is limited."

14) L125: Instead of using site numbers, use ecologically meaningful descriptors, e.g., "high-inundation site."

*Thank you for the valuable addition. We have changed the site numbers to the % marsh loss for each site as follow:*

*' 2% marsh loss site*

*11 % marsh loss site*

*33 % marsh loss site*

*58 % marsh loss site*

*lower elevation site*

*bare patches site'*

15) L128: Clarify the phrase "5 in each of four categories". This was initially unclear. Introduce categories earlier.

*See reply to comment 16 below.*

16) L134-135: Remove sentence about north/south pond sampling. It's not in Fig. 1B and feels out of context. This should be explained later when reporting on data collection.

*Reply to 15 and 16 combined: Thank you for two valuable suggestions. We have changed the order to first specify the categories of ponds and then state how much ponds of each category were sampled. We do believe that stating the north and south sampling at the ponds fits in the section of the sampling design, but we have removed the reference to Fig. 1b as it indeed might not be very clear there:*

Line 135-143: "Additionally, we  categorized ponds into four pond classes (Fig. 1b), ... Five ponds of each category were selected for sampling and for each pond, the north and south side was sampled."

17) L150: Clarify what "marsh point" means. Was there one measurement per location (n=20), or five per location (n=100)? Specify.

*You are correct that it isn't very clear this way. We have added a clarification of which points are meant between brackets:*

Line 158: "At each  marsh point (five plots in the 2 %, 11 % and 33 % marsh loss site and 17 in the 58 % marsh loss site),..."

18) L158: You sampled soil strength to 30 cm but only harvested biomass to 15 cm. Explain this methodological choice.

*Thank you for this suggestion. The shear vane soil strength was measured at the top of the profile (0-10 cm) to investigate the relationship between soil shear strength and the amount of belowground biomass. The 30 cm measurement was assumed to be below the rooting depth of Schoenoplectus americanus, as to try and see the effect of belowground biomass on the soil shear strength. This assumption has also been verified in a field campaign for a different study in this area, where we noticed roots were hardly present anymore at deeper depths. We have also clarified this in the methodology.*

Line 159-160: "...we measured the shear vane soil strength just below the soil surface (within the rooting zone) and at 30 cm below the soil surface (below the  rooting zone)

19) L162: Again, clarify what "each point" refers to (see Comment 17).

*Thank you for the suggestion. We have added a clarification of which points are meant between brackets:*

Line 169: "At the marsh locations (five plots in the 2 %, 11 % and 33 % marsh loss site and 10 in the 58 % marsh loss site ), soil cores..."

Line 174: "At each point (five plots in the 2 %, 11 % and 33 % marsh loss site and 17 in the 58 % marsh loss site),..."

20) L174: Use consistent past tense throughout the Results.

*Thank you for noticing our inconsistency. The results section has been adapted to be in past tense.*

21) L170-171: Briefly describe what red, white rhizomes, etc., are rather than only citing a source. What do they signify ecologically?

*Thank you for the valuable suggestion. We have added this sentence to specify a bit more what the ecological significance is:*

Line 184-186: "The different biomass fractions are characterised by differences in chemical composition (e.g. lignin content and C/N ratio), which has an effect on the decomposition rate (Saunders et al., 2006; Scheffer & Aerts, 2000)."

22) L177-178 and Table 1: Clarify how hydroperiod (% inundation) was measured and whether values vary within sites. Add standard deviations for variables with within-site variation, as in Table 2. Also, this statement cites Fig. 2, which shows biomass, not hydroperiod, so the reference may be misplaced.

*In the method section, there is a part on the water level time series:*

*Line 150-152: "Further, we calculated for each sampling location the duration of tidal inundation (further referred to as the hydroperiod) as the % of time that the water level is higher than the soil surface elevation of the location (Table 1)."*

*We have added the standard deviations of the elevation and hydroperiod in the table.*

Table 1: Overview of properties of the field sampling locations (Fig. 1): number of samples per location, mean surface elevation (m above local mean sea-level (m amsl)), tidal range (m), and hydroperiod (% of time that a location is inundated by tides). The numbers in the pond location categories refer to the average diameter of the ponds.

| Sampling location | Vegetation present? | Number of locations (n) | Mean elevation (m amsl) | Hydro-period (%) | Mean tidal range (m) |
|---|---|---|---|---|---|
| Marsh locations: | | | | | |
| 2% marsh loss site | Yes | 5 | 0.35±0.006 | 29.4±0.82 | 0.63 |
| 11 % marsh loss site | Yes | 5 | 0.16±0.007 | 54.3±1.43 | 0.31 |
| 33 % marsh loss site | Yes | 5 | 0.12±0.005 | 58.2±1.60 | 0.20 |
| 58 % marsh loss site | Yes | 5 | 0.11±0.002 | 73.7±0.93 | 0.06 |
| , Lower elevation site | Yes | 5 | 0.07±0.014 | 86.5±3.66 | 0.06 |
| Bare patches site | No | 7 | 0.04±0.031 | 91.7±5.29 | 0.06 |
| Pond locations: | | | | | |
| <10 m, unconnected ponds | No | 10 | -0.06±0.027 | 100 | 0.06 |
| 10-20 m, unconnected ponds | No | 10 | -0.08±0.059 | 100 | 0.06 |
| >20 m, unconnected ponds | No | 10 | -0.08±0.068 | 100 | 0.06 |
| >20 m, connected ponds | No | 10 | -0.21±0.115 | 100 | 0.06 |

*Figure 2 shows the relationship between hydroperiod and biomass on the left, so we do believe the reference here is placed correctly.*

23) L186 and L191: For L191, correlation makes sense when focusing on sites 2-4. For L186, correlation across all sites obscures the non-linear relationship (increase then decrease across hydroperiod). Suggest describing the pattern visually instead and removing the correlation. Please, see my Comment 4 as well.

*Thank you for the valuable suggestion. We have changed the text to remove the correlation and described the pattern visually and mentioned that the linear mixed models gave no significant effect:*

"Even though the regression analysis indicated no significant effect of hydroperiod on belowground biomass nor shear strength (p=0.31 and p=0.24 respectively), our  graphs seemed to indicate that the hydroperiod has  an influence on the belowground biomass  (Fig. 2a) and the shear vane soil strength (Fig. 2b) of the marsh topsoil samples (0-15 cm soil depth). There was an increase in belowground biomass and soil strength from locations at the 2 % marsh loss site (with the lowest hydroperiods around 30 %), to the 11 % marsh loss site 2 (with intermediate hydroperiods around 55 %), followed by a decrease from the 11 % marsh loss site to the lower plots of the 58 % marsh loss site (with highest hydroperiods up to >90 %). For hydroperiods ranging from 55 % up to more than 90 %, the shear vane soil strength of the topsoil decreased systematically with increasing hydroperiod  (Fig. 2b)."

24) Figures 2 and 3: Why are correlation lines and Pearson coefficients shown only in Fig. 3? Ensure consistency. Also, adjust axes to a 1:1 ratio to avoid overemphasizing y-axis variation. This may mislead interpretations (e.g., Fig. 3b).

*Thank you for the valuable suggestion. We have removed the correlation analysis for fig. 2 (see comment 23). We have removed fig3b as it was altogether based on your next comment.*

25) L206: Remove. Claiming weaker correlations requires a test for slope differences through e.g. a regression-type analysis. Pearson values alone are insufficient, and the differences are not clearly meaningful (please, see my Comment 4 as well).

*Thank you for the valuable comment. We have removed the correlation analysis on the different root types entirely and simply stated the following:*

"Additionally, we investigated whether the different root fractions had an influence on soil shear strength, but the results indicate that total root biomass rather than the biomass of individual root fractions are related to soil shear strength."

26) L215: Clarify whether p-value is exactly <0.05 or a general threshold. With such a strong correlation, p should be lower. Inconsistent p-value reporting across the manuscript (sometimes 0.0001 has been used). See Comment 3 on need for a statistical methods section.

*Thank you for the suggestion. We have added the statistical analysis as suggested in comment 3 and specified there that 0.05 is the general threshold. We have changed it throughout the manuscript so that it is consistent now.*

27) L232: Figure reference is unclear. In Fig. 5, pond and vegetated marsh soil strength seem similar (e.g., same significance letters). Yet you also refer to Fig. 3a. Clarify which figure supports which comparison.

*Thank you for the suggestion. In this paragraph we compare the results from the pond bottoms with the marsh values, both at the top soil and in the subsoil. Therefor both Fig 5 (compares the pond to the marsh subsoil) and figure 3A (shows the marsh topsoil) are necessary in this paragraph.*

28) L259: Revisit earlier comments (especially Comment 1) to ensure limitations and uncertainty are clearly reflected in the Discussion.

*Thank you for this comment. As mentioned in our reply to comment 1, we have added the limitations of the study throughout the discussion where they fit best in our opinion.*

29) L260-261: Sentence is incomplete or missing a word. Please revise.

*Thank you for noticing. As a result of one of your comments and a comment of reviewer 1, this sentence has been removed altogether.*

30) L264: Remove redundant citation – already referenced at the beginning of the sentence.

*Thank you for noticing, the citation has been removed at the end of the sentence.*

31) L300: Depth-related variation affects more than just belowground biomass. Ensure other variables are considered in the limitations paragraph (see Comment 1).

*Thank you for this very valuable comment. We have added a paragraph in the discussion going into more detail on other potential influencing factors.*

Line 362-379: "We recognise that other factors, which are not considered in our study, may influence vertical variations in soil strength. For instance, higher water content has been shown to decrease the soil penetration resistance (Gillen et al., 2021; Stoorvogel et al., 2025). As soil water content may be higher in deeper soil layers, this may also contribute to lower soil strength deeper in the profile. Yet we expect this plays a minor role as field observations typically indicate water saturated soils over the whole soil profile. Additionally, variations in soil strength along the spatial marsh degradation gradient may be related to factors we did not account for. For instance, higher nutrient loading has been shown to decrease the soil organic matter content and belowground vegetation biomass and has been reported to be related to reduced soil strength

(Turner et al., 2020). Bioturbation, especially burrowing by crabs, can increase the oxygenation of the sediment and facilitate the breakdown of belowground biomass (Wilson et al., 2012). Yet we have no data to test whether such factors varied along the spatial marsh degradation gradient and if they contributed to the observed spatial pattern of decreasing soil strength with increasing marsh degradation. Lastly, sediment properties such as organic matter content, bulk density and clay content may play a role in the cohesion of sediment (Feagin et al., 2009; Gillen et al., 2021; Joensuu et al., 2018). Higher organic matter content may increase the sediment erosion resistance, which corresponds to our finding of higher organic matter content in the sites with higher shear and penetration resistance. Studies have shown that both higher bulk density and clay content decrease the erodibility of the marsh sediment (Brooks et al., 2022; Feagin et al., 2009b; Gillen et al., 2021; Lo et al., 2017; Stoorvogel et al., 2025). These studies are however located in minerogenic marsh systems, where bulk densities and clay contents are generally higher than in organogenic systems as ours. Therefor we believe that the influence of belowground biomass on shear and penetration resistance will dominate over the effect of bulk density and clay content.

32) L305-308: Define jargon (e.g., "undercutting," "cantilever failures") or replace with simpler language. Ensure accessibility to non-specialist readers.

*Thank you for this valuable suggestion. We added the definition of both terms in the text between brackets:*

Line 380-385: "Moreover, once ponds are formed, we may expect that the marsh edges surrounding the ponds are vulnerable to increased erodibility of the exposed weaker subsoil, which may promote undercutting of the rooted top layer (i.e. erosion of the deeper subsoil) and subsequent cantilever failures (i.e. when the topsoil block remaining after undercutting collapses), a mechanism that is found to be important in driving lateral erosion of scarped marsh edges with undercutting (Bendoni et al., 2016).

---

## Author Response (AR1)

**Response letter to reviewers – Huyzentruyt et al. 2025**

**Reviewer 1**

Schepers et al. look to link tidal inundation from sea-level rise to decreasing soil strength in salt marshes, which can be related to the loss of belowground biomass from the increasing tidal inundation. The findings of this paper make sense based on what we know from previous work; however, the methods used here have issues/limitations that need to be addressed. There are also other variables that could impact soil strength in these marshes that are not discussed. Additionally, I believe the authors undervalue previous marsh work in the investigation of tidal inundation on soil strength.

*We would like to thank the reviewer for the very insightful and critical comments on our manuscript. We have tried to include them in the manuscript.*

Major Comments:

- Much of the paper's discussion and findings is based on the use of a shear vane device to interpret soil stability. This includes findings that belowground biomass increases soil strength. However, I think the authors need to acknowledge the issues related to using a shear vane as an indicator of soil stability, including that roots within the soils will directly impact the vane measurements. Hence it would make sense that there is a relationship between belowground biomass and soil/root strength if roots impact these measurements. Vane measurements can also be impacted by how fast the vane is rotated, which can vary across users.

  *Thank you for this valuable comment. We have added a part in the discussion highlighting that a relation between shear strength and belowground biomass is expected due to the methodology:*

  *"Our first main finding is the increase in marsh shear strength (Fig. 3b) and penetration resistance (see Appendix, Fig A2) with increasing belowground vegetation biomass. This can be partly explained by the methodological choice of using a shear vane for soil strength measurements, since there is of course a direct interaction between the vane and the roots, making the positive relation found between biomass shear strength a very logical one (Brooks et al., 2023)."*

  *Additionally, as we were aware of the differences that can occur between different people taking the measurements, it was assured that they were taken by the person each time while conducting the fieldwork.*

- The paper also is missing context as to what the differences in soil strength really mean between sites. From a soil strength perspective, is the difference in strengths depicted in Figure 3 and Figure 5 really large enough to generate significant differences in the erodibility of the soil? And how do these differences compare to other studies? Comparing Figure 3 to the penetrologger data in Figure 4, the penetrologger differences are much

larger than the vane differences so how does this fit into the narrative? The penetrologger data is not used in this paper as much as the vane data.

*Thank you for this comment. We have split this answer into different points to respond to each question separately.*

*From a soil strength perspective, is the difference in strengths depicted in Figure 3 and Figure 5 really large enough to generate significant differences in the erodibility of the soil?*

- *Since the shear strength ranges from >60 to less than 10 $10^3 N/m^2$ and other studies hardly ever report values above 60, we do believe that this shows a significant decrease in shear strength. Additionally, if you go to these field sites, the different in soil stability are immediately felt. The sites with the highest shear strength values are very accessible and walkable, while the lowest shear strength sites are very broken up and if you take a wrong step you can sink about 50 cm into the sediment. We have also found a paper where they measure both the erosion and the shear strength of two marsh systems and their results show that even relatively small differences (smaller than the ones we observe here) correspond with differences in erosion rates.* We have added this part to the discussion to highlight the link between erosion and shear strength.

  Line 389-392: "Our results also indicate that pond bottoms have  weak soils. Based on the findings of Stoorvogel, Willemsen, et al. (2025), where both shear strength and erosion were studied, that even relatively small differences in shear strength can correspond with large differences in erosion rates, we assume that our pond bottoms are very vulnerable to erosion."

*How do these differences compare to other studies?*

- *According to a recent paper by Brooks et al. 2023 the direct comparison of shear strength values between studies is difficult, since often different tools are used that can significantly influence the absolute values. Additionally, as you mentioned in a previous comment, the shear vane measurements are impacted by the person performing the measurements. Therefor we have chosen not to directly our values with other values, but focussing on the relative differences we see (and which are confirmed by other studies). However we did check whether our shear vane values were in the same range as other publications, which is the case.*

- Comparing Figure 3 to the penetrologger data in Figure 4, the penetrologger differences are much larger than the vane differences so how does this fit into the narrative? The penetrologger data is not used in this paper as much as the vane data.

  - *The penentrologger measured the resistance of the soil to penetration, while the shear vane measured the shear resistance of the soil. These are two different measures of sediment strength and two different measurement methods, so differences between them are logical. However, the trends we see in the shear vane data, like the decrease in strength with depth, are also observed in the penetrologger data. We have integrated the penetrologger data more in the discussion.*

- For the penetrologger profiles in Figure 4, which have the max strength near the surface, are there other factors other than roots that can also contribute to these differences along depth? For example, how does soil moisture change downcore? You only provide soil moisture data for the topsoil. Where is the BD and OM data? Also, although you may no longer have live roots at deeper depths, you would expect to see dead roots along the profile that should also impact soil strength.

*Thank you for this very accurate question. Water content, bulk density and organic matter were only analysed in the top 15 cm. However, data from a more recent study in the Blackwater marshes (Huyzentruyt et al., in review) show that the bulk density is fairly constant over a depth of 40 cm. There is however some variation of organic carbon with depth, but without a clear trend along the gradient of increasing hydroperiod (sometimes there is an increase with depth, other times a decrease). This does indeed point to the presence of dead roots at depth, which could indeed influence the strength, but we do believe that this effect on the penetration resistance is lower than for living roots (giving the lack of turgor pressure for example). We have added some additional discussion on these points (see also comment below).*

- There are other variables not mentioned by the authors that could impact soil strength rather than just belowground biomass, including potential differences in bioturbation and differences in grain size distributions. There are also other factors other than tidal inundation that impact belowground biomass, including nutrient loadings, which can vary even along a marsh site.

*Thank you for this valuable comment. We have added a paragraph to the discussion integrating more potential influences on soil strength:*

"We recognise that other factors, which are not considered in our study, could influence downward soil strength. First, higher water content has been show to decrease the soil penetration resistance (Gillen et al., 2021; Stoorvogel et al., 2025). As soil drainage is limited further away from the creek (Ursino et al., 2004; Van Putte et al., 2020) and closer to the permanently submerged ponds, we can expect that this also contributes to lower soil strength deeper in the profile. Additionally, factors such as nutrient loading (Turner et al., 2020) and bioturbation (Wilson et al., 2012) have been shown to influence belowground biomass. Higher nutrient loading has been shown to decrease the organic matter content and belowground biomass of vegetation (Turner et al., 2020). Bioturbation, especially burrowing by crabs can increase the oxygenation of the sediment and facilitate the breakdown of belowground biomass (Wilson et al., 2012). Lastly, sediment properties such as organic matter, bulk density and clay content play a role in the cohesion of sediment (Feagin et al., 2009; Gillen et al., 2021; Joensuu et al., 2018). Higher organic matter content increases the sediment erosion resistance, which corresponds to our finding for topsoil organic matter, with higher organic matter content in the sites with higher shear and penetration resistance. Studies have shown that both higher bulk density and clay content decrease the erodibility of the marsh sediment (Brooks et al., 2022; Feagin et al., 2009b; Gillen et al., 2021; Lo et al., 2017; Stoorvogel et al., 2025), these studies are however located in minerogenic marsh systems, where bulk densities and clay contents are generally higher than in organogenic systems as ours. Therefor we believe that the influence of belowground biomass on shear and penetration resistance will dominate over the effect of bulk density and clay content."

- Data needs to be made publicly available either through supplemental materials or a data repository before this paper can be published.

  *You are correct. We are working on compiling the data, so that by the time of publication a public repository will exist with the data.*

Minor Comments:

- Sea-level rise should be hyphenated throughout.

  *We have adjusted the text so that sea-level rise is always hyphenated*

- Figure 1 can be improved by 1) making the symbol size larger (the symbols in 1b are too small); 2) making some of the text larger, including the labels of a, b, and c in the Figures; 3) adding in a symbol to indicate direction (e.g. a north arrow); 4) correcting the a,b,c labels in the figure caption to be consistent with the labeling in the figures. Also check with the journal requirements to see what type of labeling is required.

  *Thank you for the suggestions. The changes have been made to figure 1.*

- Line 107 – yr-1 should be made a superscript and applied throughout the rest of the manuscript.

  *Thank you for noticing, it has now been changed accordingly*

- Also line 107 – Has the rate of SLR changed over time?

  *Thank you for this question. Yes, the sea level has increased over time. (source: NOAA tides and currents data)*

[Figure]

- Section 2.3. – Later you bring up measurements down to 80 cm but you don't mention this anywhere in this section.

  *Thank you for noticing, we have added a more detailed description in section 2.3.*

  "The measurement was taken at all marsh (five plots in the 2 %, 11 % and 33 % marsh loss site and 17 in the 58 % marsh loss site) an pond sites in the upper 80 cm of sediment."

- Section 2.4. heading should be changed to include soil C

  *Thank you for the suggestion, we have changed the title of Section 2.4 to:*

*"Belowground biomass sampling and sediment analysis"*

- Why only collect soils to 15cm depth when you measure soil strength to 30cm?

  *Thank you for this question, as this is indeed not clear from the text. The shear vane soil strength was measured at the top of the profile (0-10 cm) to investigate the relationship between soil shear strength and the amount of belowground biomass. The 30 cm measurement was assumed to be below the rooting depth of Schoenoplectus americanus, as to try and see the effect of belowground biomass on the soil shear strength. This assumption has also been verified in a field campaign for a different study in this area, where we noticed living roots were hardly present anymore at deeper depths.*

- How much time was soil ashed at 550 to determine organic content?

  *We have added more specifications on the LOI protocol in the methods as follows:*

  "One half was dried, ground and homogenized with a 0.5-mm grinder (Retsch ZM2000) and heated to 550°C and ashed for four hours to determine the organic content of the soil samples (loss on ignition)."

- Organic content is barely mentioned in the manuscript

  *Thank you for the valuable comment. We have extended the analysis on organic matter by looking at differences between sites and including it in the linear mixed model for shear strength. In correspondence with comments from you and reviewer 2, we have added a paragraph describing other potential influencing factors, among which organic matter. (see comment 2).*

- Table 1 – is the font in the last line of the caption different from the rest of the caption? It looks smaller to me.

  *Thank you for noticing, we have corrected the font size so that it matches.*

- Table 2 – font size of this table looks different from the other tables.

  *Thank you for noticing, we have corrected the font size so that it matches.*

- Figure 3 – the gray and yellow coefficients are hard to read and all coefficients can be made larger.

  *Thank you for the suggestion. In accordance with the comments of reviewer 2, we have changed the content of fig 3 (fig 3A became fig 3b and 3b was scrapped). Additionally, we increased the size and darkened the color of the coefficient in the previous figure 3a according to your comment.*

- Line 223 – unclear what difference the authors are referring to.

  *Thank you for the valuable suggestion. We have changed the phrase as follows:*

  "At soil depths below 30 cm this  variability between sites was not systematically present anymore (Fig. 4)."

- Figure 5 – Why are you comparing pond soils and bare patches to just the 30cm marsh depth rather than the 15cm marsh depth?

*Thank you for this question. Since pond bottoms and bare patches don't contain living vegetation, we can also assume that there hardly any living belowground biomass left. Therefor it seems more logical for the figure to compare these values with the 30 cm depth values, as they were taken to represent densities below the rooting depth of Schoenoplectus americanus. In the text however, we do also compare the results to the top 15 cm marsh results shown in figure 3.*

- Line 260 – Do studies that examine soil strength with tidal inundation along a marsh elevation gradient not count?

*Thank you for the question. We have emphasised the difference between our and other studies a bit more:*

"Our study is to our knowledge the first providing direct empirical evidence of the relationships between sea-level rise induced increasing tidal inundation (induced by sea level rise), decreasing soil strength, and increasing marsh to pond conversion. While we do acknowledge that the observational nature of the study complicates a generalisation of the causal relationships we found, this does not take away that the patterns that we observe are there. Moreover our findings are confirmed by similar studies, Although no previous studies a field gradient of increasing marsh to pond conversion exist, there are recent studies that demonstrate relationships between marsh soil strength and tidal hydroperiod, based on marsh locations along a gradient from low to high marsh. For instance, Jafari et al. (2024) and Stoorvogel et al. (2024; 2025) found a decrease in marsh soil strength with increasing tidal hydroperiod along a field gradient from low to high marsh locations."

- Line 261 – seems like a word or two is missing here.

*Thank you for noticing. Based on the previous comment and a comment of reviewer 2 this sentence is now changed and the issue of the missing words is resolved.*

**Reviewer 2**

In their manuscript *Sea level rise in a coastal marsh: linking increasing tidal inundation, decreasing soil strength and increasing pond expansion*, Schepers et al. highlight an interesting mechanism – previously demonstrated experimentally – by which sea level rise and associated increases in tidal inundation may promote pond expansion and reduce vegetated marsh area via declines in soil strength, likely driven by reduced belowground biomass. I found the manuscript engaging and generally well written. However, I have several major and minor comments that I strongly encourage the authors to address. In particular, the observational nature of the study – and its inherent limitations – along with the analytical approach, require clearer justification and discussion.

*We would like to thank the reviewer for the very insightful and critical comments on our manuscript. We have tried to include them in the manuscript.*

**MAJOR COMMENTS**

1) The study is observational and conducted in a specific system (microtidal marsh with organic-rich soils), which limits causal inference. This is important to emphasize when comparing with other marsh types and when interpreting findings. Please add a paragraph in the Discussion

outlining the study's limitations and potentially suggesting next steps. For instance, in L121-122, while site-specific elevation is acceptable, elevation-driven variation may influence soil strength. Unlike experimental approaches that isolate variables, your study interprets a natural gradient with inherent co-variation. This is valuable, but should be framed as such.

*Thank you for the valuable comment. We have reframed part of our discussion to highlight the observational nature of our study:*

"Our study is to our knowledge the first providing direct empirical evidence of the relationships between sea-level rise induced increasing tidal inundation, decreasing soil strength, and increasing marsh to pond conversion. While we do acknowledge that the observational nature of the study complicates a generalisation of the causal relationships we found, this does not take away that the patterns that we observe are there. Moreover, our findings are confirmed by similar studies,  based on marsh locations along a gradient from low to high marsh. For instance, Jafari et al. (2024) and Stoorvogel et al. (2024; 2025) found a decrease in marsh soil strength with increasing tidal hydroperiod along a field gradient from low to high marsh locations."

*We have made the decision to not add a separate paragraph on the limitations of the study, as we believe this undermines the strength of a discussion. Instead we have added the limitations you mentioned in the text where they apply.*

2) Your replication is at the site level, but the 5 locations within sites are treated as independent replicates. Thus, your statistical inference (but see Comments 3 and 4) is confounded with site. While I recognize the logistical constraints of field ecology, please acknowledge this in your limitations section (see Comment 1) and clarify the implications for interpreting your results.

*Thank you for the valuable advise. We have changed our statistical analysis to linear mixed models, to include in the random effect of site. See also response to comment 3 and 4.*

3) There is no dedicated paragraph describing your statistical analysis. Readers need a clear overview of how hypotheses were tested, which variables were used, whether data were averaged, and which models were applied. Please add a paragraph detailing your statistical approach.

*Thank you for the very true suggestion. We have added a paragraph on the statistics:*

"2.5 Statistical analysis

The effect of hydroperiod on shear strength and belowground biomass was analysed using linear mixed models (LMM), using field site as a random effect to account for within site clustering. A separate LMM analysis was performed to evaluate the influence of organic matter content, bulk density, water content, hydroperiod and belowground biomass on shear strength, again incorporating field site as a random effect. The differences in bulk density, water content, organic matter, shear strength and belowground biomass between sites were analysed using pairwise Wilcoxon rank sum test with Bonferroni correction. All analyses were executed in R (R core team, 2022), using the lme4 package (Bates et al., 2015) for the linear mixed models. The p-value threshold used is 0.05. "

4) Related to the above: while correlation may be suitable for Fig. 3, hydroperiod is unlikely to be a response variable influenced by belowground biomass or soil strength. Therefore, regression would be more appropriate to suggest directional relationships in Fig. 2. If you intentionally chose

correlation, please explain why. See also Comment 3 regarding the missing statistical analysis section.

*Thank you for the advise. We have changed the statistical analysis from only correlation testing to also include linear mixed models (see also response above).*

**RELATIVELY MINOR COMMENTS**

5) L14: Replace "method" with "mechanism"; rephrase the sentence accordingly.

*Thank you for the suggestion. We have changed the word.*

"Here, we propose another mechanism..."

6) L14 and L51: Define "soil strength" clearly at first appearance in both the abstract and main text.

*Thank you for the valuable suggestion. We have added a more detailed explanation on what is meant by "soil strength" both in the abstract and the main text.*

*Abstract:* "Here, we propose another mechanism between sea-level rise, increasing marsh inundation, and decreasing marsh soil resistance to shear and penetration stress (further mentioned as soil strength),..."

*Main text:* "...we investigate the hypothesis that the marsh soil strength (measured as resistance against shear and penetration stress)..."

7) L69-70: Clarify what is meant by "stable marsh system." Do you mean a system not subject to sea level rise?

*Thank you for the suggestion. We have changed the text as such:*

"This relationship was however quantified in a marsh system without signs of degradation as a result of sea level rise"

8) L70: The phrasing suggests a direct link between soil strength, marsh loss, erosion, and pond expansion. Please revise to reflect the uncertainty of these associations.

*Thank you for the suggestion. We have changed the sentence as such:*

"Our analysis  suggests relationships between..."

9) L72: "Microtidal" should be hyphenated or not consistently throughout. Ensure consistency in terminology across the manuscript.

*Thank you for noticing. The non-hyphenated version has been used throughout the whole text.*

10) L82-84: Rephrase this section to introduce the experimental design first (number of sites, selection criteria) before referring to site numbers.

*Thank you for the valuable suggestion. We have removed the mention of site numbers here all together:*

"The mean tidal range decreases from 63 cm at Fishing Bay (bottom right of Fig. 1a)  to 6 cm at Lake Blackwater (top left of Fig. 1a)"

11) L88-93 (Figure caption): (1) Capitalization of letters should match figure; (2) use "panel" instead of "figure" for subplots; (3) explain the inset; (4) clarify data source for marsh loss values.

*Thank you for the suggestion. The caption has been changed accordingly.*

**"Figure 1: a: Aerial images of the Blackwater marshes (black: water, light grey: marsh) with sampling locations (Copernicus – Sentinel data [2025]. Retrieved from Google Earth Engine, processed by ESA). The marsh loss (i.e. proportion of shallow open water ponds to total marsh area) is quantified for each site as 2 % for site 1, 11 % for site 2, 33 % for site 3 and 58 % for site 4** **(based on Schepers et al. (2017)). The inset map shows the location of the Blackwater marshes in the Chesapeake Bay****. The green box is the extent of** **panel b. b:** **pond locations (white) sampled at site 4. Values in the legend of** **(b)** **refer to the average pond diameter in each category. The yellow box is the extent of** **panel c. c:** **marsh locations at site 4 with (green) and without (yellow) vegetation."**

12) L97-98: If you use cardinal directions, mark them on the figure. Otherwise, use terms like "left" or "right."

*We added a northern arrow to the map to resolve this issue.*

13) L104: Acknowledge limitations of space-for-time substitution briefly here and expand in Discussion (see Comment 1).

*Thank you for this valuable comment. We have added some nuance on the space-for-time substitution in the discussion:*

*"Of course, since we are using a space-for-time substitution, there could be other differences between sites (such as salinity and tidal range) that could influence the vegetation belowground biomass production, however given the agreement of our results with these previous findings, we believe that this effect is limited."*

14) L125: Instead of using site numbers, use ecologically meaningful descriptors, e.g., "high-inundation site."

*Thank you for the valuable addition. We have changed the site numbers to the % marsh loss for each site as follow:*

*' 2% marsh loss site*

*11 % marsh loss site*

*33 % marsh loss site*

*58 % marsh loss site*

*lower elevation site*

*bare patches site'*

15) L128: Clarify the phrase "5 in each of four categories". This was initially unclear. Introduce categories earlier.

16) L134-135: Remove sentence about north/south pond sampling. It's not in Fig. 1B and feels out of context. This should be explained later when reporting on data collection.

*15 and 16 combined: Thank you for two valuable suggestions. We have changed the order to first specify the categories of ponds and then state how much ponds of each categories were sampled. We do believe that stating the north and south sampling at the ponds fits in the section of the*

*sampling design, but we have removed the reference to Fig. 1b as it indeed might not be very clear there:*

"Additionally, we  categorized ponds into four pond classes (Fig. 1b), ... Five ponds of each category were selected for sampling and each pond, the north and south side was sampled."

17) L150: Clarify what "marsh point" means. Was there one measurement per location (n=20), or five per location (n=100)? Specify.

*You are correct that it isn't very clear this way. We have added a clarification of which points are meant between brackets:*

"At each point (five plots in the 2 %, 11 % and 33 % marsh loss site and 17 in the 58 % marsh loss site),..."

18) L158: You sampled soil strength to 30 cm but only harvested biomass to 15 cm. Explain this methodological choice.

*Thank you for this suggestion. The shear vane soil strength was measured at the top of the profile (0-10 cm) to investigate the relationship between soil shear strength and the amount of belowground biomass. The 30 cm measurement was assumed to be below the rooting depth of Schoenoplectus americanus, as to try and see the effect of belowground biomass on the soil shear strength. This assumption has also been verified in a field campaign for a different study in this area, where we noticed roots were hardly present anymore at deeper depths. We have also clarified this in the methodology.*

19) L162: Again, clarify what "each point" refers to (see Comment 17).

*Thank you for the suggestion. We have added a clarification of which points are meant between brackets:*

"At each point (five plots in the 2 %, 11 % and 33 % marsh loss site and 17 in the 58 % marsh loss site),..."

20) L174: Use consistent past tense throughout the Results.

*Thank you for noticing our inconsistency. The results section has been adapted to be in past tense.*

21) L170-171: Briefly describe what red, white rhizomes, etc., are rather than only citing a source. What do they signify ecologically?

*Thank you for the valuable suggestion. We have added this sentence to specify a bit more what the ecological significance is:*

"The different biomass fractions are characterised by differences in chemical composition (e.g. lignin content and C/N ratio), which has an effect on the decomposition rate (Saunders et al., 2006; Scheffer & Aerts, 2000)."

22) L177-178 and Table 1: Clarify how hydroperiod (% inundation) was measured and whether values vary within sites. Add standard deviations for variables with within-site variation, as in Table 2. Also, this statement cites Fig. 2, which shows biomass, not hydroperiod, so the reference may be misplaced.

*In the method section, there is a part on the water level time series "Further, we calculated for each sampling location the duration of tidal inundation (further referred to as the hydroperiod) as the % of time that the water level is higher than the soil surface elevation of the location (Table 1)." We have added the standard deviations of the elevation and hydroperiod in the table.*

*Figure 2 shows the relationship between hydroperiod and biomass on the left, so we do believe the reference here is placed correctly.*

23) L186 and L191: For L191, correlation makes sense when focusing on sites 2-4. For L186, correlation across all sites obscures the non-linear relationship (increase then decrease across hydroperiod). Suggest describing the pattern visually instead and removing the correlation. Please, see my Comment 4 as well.

*Thank you for the valuable suggestion. We have changed the text to remove the correlation and described the pattern visually and mentioned that the linear mixed models gave no significant effect:*

"Even though the regression analysis indicated no significant effect of hydroperiod on belowground biomass nor shear strength (p=0.31 and p=0.24 respectively), our  graphs seemed to indicate that the hydroperiod has  an influence on the belowground biomass  (Fig. 2a) and the shear vane soil strength hydroperiod hydroperiod (Fig. 2b) of the marsh topsoil samples (0-15 cm soil depth). There was an increase in belowground biomass and soil strength from locations at the 2 % marsh loss site (with the lowest hydroperiods around 30 %), to the 11 % marsh loss site 2 (with intermediate hydroperiods around 55 %), followed by a decrease from the 11 % marsh loss site to the lower plots of the 58 % marsh loss site (with highest hydroperiods up to >90 %). For hydroperiods ranging from 55 % up to more than 90 %, the shear vane soil strength of the topsoil decreased systematically with increasing hydroperiod  (Fig. 2b)."

24) Figures 2 and 3: Why are correlation lines and Pearson coefficients shown only in Fig. 3? Ensure consistency. Also, adjust axes to a 1:1 ratio to avoid overemphasizing y-axis variation. This may mislead interpretations (e.g., Fig. 3b).

*Thank you for the valuable suggestion. We have removed the correlation analysis for fig. 2 (see comment 23). We have removed fig3b as it was altogether based on your next comment.*

25) L206: Remove. Claiming weaker correlations requires a test for slope differences through e.g. a regression-type analysis. Pearson values alone are insufficient, and the differences are not clearly meaningful (please, see my Comment 4 as well).

*Thank you for the valuable comment. We have removed the correlation analysis on the different root types entirely and simply stated the following:*

"Additionally, we investigated whether the different root had an influence on soil shear strength, but it seems that quantity rather than quality is important."

26) L215: Clarify whether p-value is exactly <0.05 or a general threshold. With such a strong correlation, p should be lower. Inconsistent p-value reporting across the manuscript (sometimes 0.0001 has been used). See Comment 3 on need for a statistical methods section.

*Thank you for the suggestion. We have added the statistical analysis as suggested in comment 3 and specified there that 0.05 is the general threshold. We have changed it throughout the manuscript so that it now is consistent.*

27) L232: Figure reference is unclear. In Fig. 5, pond and vegetated marsh soil strength seem similar (e.g., same significance letters). Yet you also refer to Fig. 3a. Clarify which figure supports which comparison.

*Thank you for the suggestion. In this paragraph we compare the results from the pond bottoms with the marsh values, both at the top soil and in the subsoil. Therefor both Fig 5 (compares the pond to the marsh subsoil) and figure 3A (shows the marsh topsoil) are necessary in this paragraph.*

28) L259: Revisit earlier comments (especially Comment 1) to ensure limitations and uncertainty are clearly reflected in the Discussion.

*Thank you for this comment. As mentioned on comment 1, we have added the limitations of the study throughout the discussion where they fit best in our opinion.*

29) L260-261: Sentence is incomplete or missing a word. Please revise.

*Thank you for noticing. As a result of one of your comments and a comment of reviewer 1, this sentence has been removed altogether.*

30) L264: Remove redundant citation – already referenced at the beginning of the sentence.

*Thank you for noticing, the citation has been removed at the end of the sentence.*

31) L300: Depth-related variation affects more than just belowground biomass. Ensure other variables are considered in the limitations paragraph (see Comment 1).

*Thank you for this very valuable comment. We have added a paragraph in the discussion going into more detail on other potential influencing factors.*

"We recognise that other factors, which are not considered in our study, could influence downward soil strength. First, higher water content has been show to decrease the soil penetration resistance (Gillen et al., 2021; Stoorvogel et al., 2025). As soil drainage is limited further away from the creek (Ursino et al., 2004; Van Putte et al., 2020) and closer to the permanently submerged ponds, we can expect that this also contributes to lower soil strength deeper in the profile. Additionally, factors such as nutrient loading (Turner et al., 2020) and bioturbation (Wilson et al., 2012) have been shown to influence belowground biomass. Higher nutrient loading has been shown to decrease the organic matter content and belowground biomass of vegetation  (Turner et al., 2020). Bioturbation, especially burrowing by crabs can increase the oxygenation of the sediment and facilitate the breakdown of belowground biomass (Wilson et al., 2012). Lastly, sediment properties such as organic matter, bulk density and clay content play a role in the cohesion of sediment (Feagin et al., 2009; Gillen et al., 2021; Joensuu et al., 2018). Higher organic matter content increases the sediment erosion resistance, which corresponds to our finding for topsoil organic matter, with higher organic matter content in the sites with higher shear and penetration resistance. Studies have shown that both higher bulk density and clay content decrease the erodibility of the marsh sediment (Brooks et al., 2022; Feagin et al., 2009b; Gillen et al., 2021; Lo et al., 2017; Stoorvogel et al., 2025), these studies are however located in minerogenic marsh systems, where bulk densities and clay contents are generally higher than in organogenic systems as ours. Therefor we believe that the influence of belowground biomass on shear and penetration resistance will dominate over the effect of bulk density and clay content."

32) L305-308: Define jargon (e.g., "undercutting," "cantilever failures") or replace with simpler language. Ensure accessibility to non-specialist readers.

*Thank you for this valuable suggestion. We added the definition of both terms in the text between brackets:*

"Moreover, once ponds are formed, we may expect that the marsh edges surrounding the ponds are vulnerable to increased erodibility of the exposed weaker subsoil, which may promote undercutting (i.e. erosion of the subsoil layer) of the rooted top layer and subsequent cantilever failures (i.e. when the topsoil block remaining after undercutting collapses), a mechanism that is found to be important in driving lateral erosion of scarped marsh edges with undercutting (Bendoni et al., 2016).

---

## Author Response (AR2)

**Public justification (visible to the public if the article is accepted and published)**: The paper has now been re-reviewed by one of the original reviewers, who felt that the manuscript has improved substantially. They did, however, raise one remaining concern that I also share. Specifically, the reviewer suggested adding a short paragraph in the discussion that addresses the study's limitations. While the authors expressed concern that such a paragraph might detract from the paper, I want to emphasize that acknowledging limitations is a hallmark of strong scientific writing. Including this section would not weaken the manuscript—it would enhance its transparency, credibility, and overall impact by helping readers understand the scope and context of the findings.

*We would like to thank the reviewer and the editor for another very thorough revision of the manuscript. We have adjusted the manuscript according to the comments.*

*Reponses to each comment are added in blue and the changed blocks of text are added, with removed text crossed out in*  *and added text just in red.*

*The main changes we made are:*

1) *Added a paragraph on the limitations of the current study*
2) *Changed figure 1, 3 and 5*
3) *Checked the text for additional spelling and lay-out errors*

Additional private note (visible to authors and reviewers only):
In addition to adding a paragraph on limitations, the reviewer also requested the following minor edits.

Major:

1. Add a paragraph addressing the limitations of your study.

We have added a separate section within the discussion to address the limitations. We therefor moved some blocks of text from the previous discussion into this section. In this response we only show the finished limitations section.

Line 415-455:
4.1 Limitations of the study
A first limitation of our study is the use of a space-for-time substitution, assuming that a spatial gradient in increasing marsh inundation and increasing pond area can be considered representative for the temporal development of increasing pond surface area within a marsh, as a result of increasing marsh inundation in response to relative sea level rise. Because of this space-for-time approach, there could be differences between sites, other than differences in inundation and pond surface area, that could influence the vegetation belowground biomass production that we have not considered. However, given the qualitative agreement of our results with previous findings who don't use this space-for-time substitution (as discussed above), we believe that this effect is limited.

Further, the use of shear vane devices is not recommended for direct comparison between different studies, as measurements are influenced by the present roots, but also the person who takes the measurements. We therefore recommend on the one hand that shear vane devices are used in combination with other methods for evaluating soil strength, such as a penetrologger (used in our study) or a Cohesive Strength Meter (Brooks et al., 2023). On the other hand, we recommend to only compare patterns and not absolute values between

studies. We argue however that when measurements are performed by the same person, shear vane measurements are valid for comparison of relative differences in sediment bed strength within a given study site, as done in our study.

Finally, we recognise that other environmental variables, which are not considered in our study, could influence vertical variations in soil strength. For instance, higher water content has been shown to decrease the soil penetration resistance (Gillen et al., 2021; Stoorvogel, de Smit, et al., 2025). As soil water content may be higher in deeper soil layers, this may also contribute to lower soil strength deeper in the profile. Yet, we expect this plays a minor role in our study sites as field observations typically indicate water saturated soils over the whole soil profile. Additionally, variations in soil strength along the spatial marsh degradation gradient may be related to factors we did not account for. For instance, higher nutrient loading decreases the soil organic matter content and belowground vegetation biomass and has been reported to be related to reduced soil strength (Turner et al., 2020). Bioturbation, especially burrowing by crabs, can increase the oxygenation of the sediment and facilitate the breakdown of belowground biomass (Wilson et al., 2012). Yet we have no data to test whether such factors varied along the spatial marsh degradation gradient and if they contributed to the observed spatial pattern of decreasing soil strength with increasing marsh degradation.

Minor:
1. Formatting of the introduction is off. Why do two of the paragraphs start with references?

R1. Something seems to have gone wrong with the references in the previous version, thank you for noticing. The references have been removed.

2. Figure 1c, the image is quite pixelated so it doesn't really add anything to the figure. Also, the symbols in 1b are still too small to read and an upward facing arrow still doesn't necessarily tell you the direction it is oriented towards.

R2. We have removed panel C and increased the size of panel B to improve the readability. The arrow is a north arrow, we have added this to the caption.

Line 89-97:

[Figure]

**Figure 1: (a): Aerial images of the Blackwater marshes (black: water, light grey: marsh) with sampling locations (Copernicus – Sentinel data [2025]. Retrieved from Google Earth Engine, processed by ESA). The marsh loss (i.e. proportion of shallow open water ponds to total marsh area) is quantified for each site based on Schepers et al. (2017). (b) Inset map showing the location of the Blackwater marshes in the Chesapeake Bay. The green box is the extent of panel  c.  (c): pond locations (white) sampled at site 4. Values in the legend of (c) refer to the average pond diameter in each category. The arrow on the bottom is a North arrow. **

3. Paragraph starting in line 110 - the authors didn't really answer my question on the changing rates of SLR at the site. The overall rate they show is 4.06 mm/yr but this encompasses a rate from 1940 to 2025. I would assume the rate of SLR is actually increasing so the timescale of the SLR rate that you state would matter to knowing whether these sites can keep up with SLR. Also, since the sedimentation study cited here is from 1985, is there any indication that sedimentation has changed over time at this site?

R3. It does indeed seem that the sea level rise rate has been increasing since the 1970's (https://tidesandcurrents.noaa.gov/sltrends/sltrends_station.shtml?plot=50yr&id=8571892). From the late '90s onwards, the sea level rise rate has exceeded the sediment accretion rates reported in the 1985 study. This is further specified in the revised text below:

Line 110-115: In particular, sediment accretion rates (on average 1.7-3.6 mm yr$^{-1}$ (Stevenson et al., 1985)) are less than the  long-term rate of relative sea-level rise of 4.06 mm yr$^{-1}$ in Cambridge, MD, calculated over the period of 1943-2025 (NOAA station 8571892, http://tidesandcurrents.noaa.gov/sltrends, 2025-04-10)), . The historical sea level rise rate has been increasing since the 1970's, and it has exceeded the sediment accretion rate since the 1990's (NOAA station 8571892, http://tidesandcurrents.noaa.gov/sltrends, 2025-04-10). Moreover, more sediment is exported from the system than imported into it (Ganju et al., 2013).

We did a more recent study in that area where we measured sediment accretion rates (the paper containing this data is currently under review) and the sedimentation rates are still around 3.6 mm/y on average in the inner marsh locations.

4. line 155 and thereafter - the authors need to be consistent with either spelling out the number of sites or using digits.

R4. This was corrected. We aimed to consistently spell out numbers below 10 as is the convention in scientific writing.

5. With the addition of linear models, while they are applicable for analyzing patterns, why not also look at using non-linear models? Only the results from below ground biomass are shown and the authors mention the other variables did not have a significant influence. But perhaps they could still be important to overall shear strength?

R5. We did in fact also test non-linear models (generalised additive models), but the results did not change, so we decided to only report on the linear models.

We compared a model with all variables to the model with only the belowground biomass and the Akaike Information Criterion is lower for the simple model than for the more extensive model. Therefor we believe that the additional variables included in the more extensive model (i.e. water content, organic matter content, bulk density) are not important in our case. We do acknowledge that previous studies have shown these variables to be important (see paragraph starting on line 344-352 in the discussion), but we also emphasize that these studies are mainly performed in minerogenic marshes, while ours is an organogenic one. In our study system, the amount of belowground biomass is the most important driver.

6. Need to fix the formatting of table 1 and some typos.
R6. We have fixed the errors in table 1.

7. table 2 caption is misplaced.

R7. Indeed, something seems to have gone wrong in the layout. It is fixed now.

8. Figure 3 caption says there are significance letters but they are missing from the actual figure. Is there enough of a sample size to make inferences of significant differences?

R8. Thank you for noticing, we seem to have looked over this error. We have changed the figure (see also comment 9) and have included error bars to indicate that every dot is the average of 5 replicate measurements. For each point we thus have 25 values, which we do believe is a big enough sample size. We have made this clearer in the caption as well.

Line 240-249:

[Figure]

**Figure 2: a) Comparison of shear soil strength (10³ N m⁻², ) in the different field locations and between the topsoil (0-10 cm, full circle) and subsoil (30-40 cm, open circle). Letters at the top show the results of the pairwise Wilcoxon rank sum test with Bonferroni correction (n=25 for each site and depth combination), with different letters indicating significant differences between sites and depths. Error bars indicate the standard deviation of each measurement (n=5 for every point) b) Total belowground biomass (kg m⁻²) versus shear vane soil strength (10³ N m⁻², for 0-10 cm soil depth) for all vegetated marsh sampling locations (no bare or pond locations), demonstrating a strong correlation (r = 0.91, p<0.05).**

9. Figure 5: You mention this in your comment to me but you can be more clear in the text for why you are comparing the ponds to the 30cm marsh depth. There is no point in the text that I can find where you directly compare the marsh pond data to the marsh 30cm data. Isn't this why you tested these for significance in the figure?

R9. You are indeed correct that in the remaining manuscript we don't compare these two. We have therefor decided to change figure 5 to only show the pond data and have added the subsoil measurements to figure 3a (see also R8). This is the new figure 5:

Line 271-276:

[Figure]

**Figure 3: Shear vane soil strength ($10^3$ N m$^{-2}$) measurements of pond topsoils (n=50 for each boxplot).  Significant differences between pond types  have different letters above each boxplot, differences between groups have different letters at the very top of the figure (pairwise Wilcoxon rank sum test with Bonferroni correction, $\alpha$= 0.05).**

10. Line 291 - only partly? Based on my comment from V1, why couldn't this be completely explained by roots impacting the shear vane measurement? Just because there is a similar relationship to biomass and penetration resistance does not mean the shear vane measurement was only partly impacted by roots.

R10. We have removed the word partly. We wanted to say that we argue there still is a causal relationship between shear strength and belowground biomass, even though the shear vane method is directly affected by roots.

Line 305-307: This can  be explained by the methodological choice of using a shear vane for soil strength measurements, since roots can be expected to directly affect the shear vane measurements (Brooks et al., 2023).

We have also added an additional paragraph in the discussion on the limitations of using shear vane measurements.

Line 421-426:
Further, the use of shear vane devices is not recommended for direct comparison between different studies, as measurements are influenced by the present roots, but also the person who takes the measurements. We therefore recommend on the one hand that shear vane devices are used in combination with other methods for evaluating soil strength, such as a

penetrologger (used in our study) or a Cohesive Strength Meter (Brooks et al., 2023). On the other hand, we recommend to only compare patterns and not absolute values between studies. We argue however that when measurements are performed by the same person, shear vane measurements are valid for comparison of relative differences in sediment bed strength within a given study site, as done in our study.

11. Line 318 - if salinity and tidal range could influence below ground biomass production, couldn't you test for this since you have biomass data and you can get salinity/tidal range data?

R11. We didn't test the effect of salinity because this is outside the scope of our study. We mention it here because we want to be complete in acknowledging that space-for-time substitutions have limitations. Therefor we mention that other factors, which we did not account for but that could differ between sites, could potentially influence the differences between sites.

Line 416-420: A first limitation of our study is the use of a space-for-time substitution, assuming that a spatial gradient in increasing marsh inundation and increasing pond area can be considered representative for the temporal development of increasing pond surface area within a marsh, as a result of increasing marsh inundation in response to relative sea level rise. Because of this space-for-time approach, there could be differences between sites, other than differences in inundation and pond surface area, that could influence the vegetation belowground biomass production that we have not considered. However, given the qualitative agreement of our results with previous findings who don't use this space-for-time substitution (as discussed above), we believe that this effect is limited.

---

## Author Response (AR3)

**Response to file upload**

Regarding your figure 1: please note that in accordance with our standards, material that originated from Google Earth requires the copyright symbol "©". Please add it to the caption of Figure 1 with the next revision. For instance, © Google Earth Engine

*We have added the copyright symbol in the caption of figure 1*.

**Figure 1: (a): Aerial images of the Blackwater marshes (black: water, light grey: marsh) with sampling locations (Copernicus – Sentinel data [2025]. Retrieved from © Google Earth Engine, processed by ESA). The marsh loss (i.e. proportion of shallow open water ponds to total marsh area) is quantified for each site based on Schepers et al. (2017). (b) Inset map showing the location of the Blackwater marshes in the Chesapeake Bay. The green box is the extent of panel c. (c): pond locations (white) sampled at site 4. Values in the legend of (c) refer to the average pond diameter in each category. The arrow on the bottom is a North arrow.**